# Hierarchical Text Classification and Its Foundations: A Review of Current Research

Alessandro Zangari [†][ID], Matteo Marcuzzo [†][ID], Matteo Rizzo [ID], Lorenzo Giudice [ID], Andrea Albarelli [ID] and Andrea Gasparetto *[ID]

Department of Environmental Sciences, Informatics and Statistics, Ca' Foscari University, 30123 Venice, Italy; alessandro.zangari@unive.it (A.Z.); matteo.marcuzzo@unive.it (M.M.); matteo.rizzo@unive.it (M.R.); lorenzo.giudice@unive.it (L.G.); albarelli@unive.it (A.A.)

* Correspondence: andrea.gasparetto@unive.it
† These authors contributed equally to this work.

**Abstract:** While collections of documents are often annotated with hierarchically structured concepts, the benefits of these structures are rarely taken into account by classification techniques. Within this context, hierarchical text classification methods are devised to take advantage of the labels' organization to boost classification performance. In this work, we aim to deliver an updated overview of the current research in this domain. We begin by defining the task and framing it within the broader text classification area, examining important shared concepts such as text representation. Then, we dive into details regarding the specific task, providing a high-level description of its traditional approaches. We then summarize recently proposed methods, highlighting their main contributions. We also provide statistics for the most commonly used datasets and describe the benefits of using evaluation metrics tailored to hierarchical settings. Finally, a selection of recent proposals is benchmarked against non-hierarchical baselines on five public domain-specific datasets. These datasets, along with our code, are made available for future research.

**Keywords:** hierarchical text classification; multilabel classification; hierarchical metrics; natural language processing; text classification survey





## 1. Introduction

Text classification (TC) is one of the most widely researched tasks within the natural language processing (NLP) community [1]. Shortly put, TC methods are supervised learning algorithms whose objective is to map documents (i.e., pieces of text) to a predefined set of labels. The most common classification setting in practice is *multiclass*, where only one label (i.e., class) can be associated with each document. There might only be two labels to choose from (binary classification) or multiple. In contrast, *multilabel* classification allows every document to be labeled with multiple categories. In either case, TC has many practical applications, such as topic labeling, sentiment analysis, and named entity recognition [2].

From a theoretical point of view, binary classification is the most generic classification scenario, as long as the categories are stochastically independent [3]. If this is the case, the problem can be translated into $|C|$ independent problems, where $C$ is a set of classes. However, there are many practical scenarios in which this is not the case; hierarchical text classification (HTC) is one of these. HTC is a sub-task of TC, as well as part of the wider hierarchical multilabel classification (HMC). Vens et al. [4] define HMC as a classification task where instances (*i*) may belong to multiple classes simultaneously, and (*ii*) are organized within a hierarchy. Thus, as labels do indeed have a dependency on each other (as made explicit by the hierarchy), the simplifying independence assumption cannot be made.

### 1.1. What Is Hierarchical Text Classification?

The distinguishing property of HTC tasks is that the data they utilize have labels organized in a multi-level hierarchy. Within this, each label can be seen as a *node* and may have many possible children. In these types of situations, the hierarchical structure is key to the achievement of well-performing classification methods.

This type of label organization is quite common; real-world classification datasets often contain a large number of categories explicitly organized into a class hierarchy or taxonomy. Moreover, this arrangement of information can sometimes be found even in corpora for which the hierarchy was not initially devised. For example, a set of documents categorized by their topics can usually be organized within large macro-areas that encompass subsets of related subjects (for instance, the "sports" macro category can contain both "tennis" and "football"). Indeed, hierarchies allow for intuitive modeling of what could instead be complex relationships among labels (such as the *is-a* and *part-of* relationships).

Thus, thanks to the increased availability of TC datasets that integrate hierarchical structure in their labels, as well as a general interest in industrial applications that utilize TC, a large number of new methods for HTC have been proposed in recent years. Indeed, HTC has many practical applications beyond classic TC, such as International Classification of Diseases (ICD) medical coding [5,6], legal document concept labeling [7], patent labeling [8], IT ticket classification [9], and more.

The strength of these "hierarchical classifiers" comes from their ability to leverage the dependency between labels to boost their classification performance. This is particularly important when considering the wider task of multilabel TC, which, in general, can be quite difficult, and even more so when dealing with large sets of labels that contain many similar or related labels [10]. Moreover, HTC methods tend to exhibit better generalization when faced with new classes compared to their non-hierarchical counterparts. Newly introduced classes are often subcategories of larger, pre-existing macro-categories. Consequently, these hierarchical methods retain some of their knowledge from the parent nodes of these newly introduced categories.

### 1.2. Related Work

Text classification is one of the most active research areas within NLP, and many surveys and reviews have been published in recent years. These cover a wide range of techniques used in NLP, from traditional methods to the latest deep learning applications [1,2,11–13]. However, these works cover the broader TC field and do not cover HTC specifically (or mention it very briefly). In the following paragraphs, we instead present some notable works within the field of HTC. We briefly touch on seminal works that describe this field, while also highlighting recently published reviews that perform an analysis of the existing methods.

One of the first works to directly address HTC is that of Koller and Sahami [14]. The authors already highlight some of the most notable characteristics of HTC, such as the inadequacy of flat classifiers as opposed to their hierarchical counterparts (which we describe in Section 3.1) and the proliferation of topic hierarchies. Sun and Lim [15] similarly address the difficulties tied to utilizing flat approaches in hierarchical settings, as well as discussing the issues related to standard performance metrics. As we will discuss in Section 3.3, classification metrics such as precision, recall, and accuracy assume independence between categories, which might give a skewed representation of a classifier's real performance (e.g., performing a misclassification on a child not but not on its parent should be considered better than entirely incorrect classifications). In a subsequent work, Sun et al. [16] propose a specification language to describe hierarchical classification methods to facilitate the description and creation of HTC systems.

More recently, Silla and Freitas [17] give a precise definition of hierarchical classification and propose a unifying framework to classify this task across different domains (i.e., not limited to text). They provide a comprehensive overview of this research area, including a conceptual and empirical comparison between different hierarchical classifica-

tion approaches, as well as some remarks on the usage of specialized evaluation metrics. Stein et al. [18], on the other hand, address HTC directly, proposing an evaluation of traditional and neural models on a hierarchical task. In particular, the authors aim to gauge the efficacy of different word embedding strategies (which we outline in Section 2.1) with several methods for the specific HTC task. Several methods are tested, also comparing the effect of different text embedding techniques by evaluating their effect on both standard and specialized metrics. Similarly to other authors, they also advocate for the inadequacy of traditional "flat" classification metrics in hierarchical settings. Lastly, Defiyanti et al. [19] provide a review of a sub-class of hierarchical methods, namely, global (big-bang) approaches (described in Section 3.1). The authors detail various algorithms using the big-bang approach, mainly focusing on applications in bioinformatics and text classification.

### 1.3. Contributions

In this work, we propose an analysis of the current research trends related to HTC, performing a systematic search of all papers that have been published in the last 5 years (i.e., between 2019 and 2023). We deem this range effective for analyzing recent work while also providing a way to limit the scope. We collect papers querying the keywords "*hierarchical text classification*", "*hierarchical multilabel*", and "*multilevel classification*" in Google Scholar (https://scholar.google.com, accessed on 17 March 2024), PapersWithCode (https://paperswithcode.com, accessed on 17 March 2024), Web of Science (https://webofscience.com, accessed on 17 March 2024), and DBLP (https://dblp.org, accessed on 17 March 2024). We complement our search results by searching with the query "*hierarchical AND 'text classification'*" on Scopus (https://www.scopus.com, accessed on 17 March 2024).

Moreover, in our experimental section, we report the performance of a set of recent proposals, as well as several baselines, on five datasets. Three of these datasets are popularly utilized in the literature, while two of them are newly proposed versions of existing collections. In summary, the main contributions of this work can be summarized as follows:

- We provide an extensive review of the state of current research regarding HTC;
- We explore the NLP background of text representation and the various neural architectures being utilized in recent research;
- We analyze HTC specifically, providing an analysis of common approaches to this paradigm and its evaluation measures;
- We summarize a considerable number of recent proposals for HTC, spanning between 2019 and 2023. Among these, we dive deeper into the discussion of several methods and how they approach the task;
- We test a set of baselines and recent proposals on five benchmark HTC datasets that are representative of five different domains of applications;
- We release our code (https://gitlab.com/distration/dsi-nlp-publib/-/tree/main/htc-survey-24, accessed on 17 March 2024) and dataset splits for public usage in research. The datasets are available on Zenodo [20], including two new benchmark datasets derived from existing collections;
- Lastly, we summarize our results and discuss current research challenges in the field.

### 1.4. Structure of the Article

The rest of this article is organized as follows. Section 2 introduces the main NLP topics and recent advancements relevant to the latest proposals in HTC. Section 3 then dives into the specifics of HTC, describing the various approaches found in the literature and hierarchical evaluation measures. Section 4 summarizes recently proposed methods, analyzing in more detail a subset of them we wish to explore in the experimental section. This section also introduces the most popular datasets utilized in HTC research. The experimental part of this survey begins in Section 5, which outlines the datasets utilized and the methods being benchmarked, as well as a discussion of the results. The manuscript

then moves towards its end in Section 6, which briefly explores current research challenges and research directions in the specific field of HTC, and draws its conclusions in Section 7.

## 2. NLP Background

Given the significant advancements in NLP over the last few years, our discussion begins with an updated overview of the NLP topics that lay the foundation of any HTC method. First, we describe text representation and classification from a generic point of view, highlighting some of the most prominent approaches to the extraction of features from text. Then, to provide sufficient background for recent methods discussed throughout this article, we highlight some of the most notable neural architectures being utilized in the literature as of now.

### 2.1. Text Representation and Classification

The interpretation of text in a numerical format is the fundamental first step of any application that processes natural language. How text is viewed and represented has changed drastically in the last few decades, moving from relatively simple statistics-based word counts to more semantically and syntactically meaningful vectorized representations [21–23]. A richer text representation translates into more meaningful features, which—if utilized adequately—lead to massive improvements in downstream task performance. In this section, we overview the main approaches to text representation, highlighting major milestones and how they differ from one another.

### 2.1.1. Text Segmentation

First, bodies of text must be segmented into meaningful units—a process called *tokenization* for historical reasons [10]. Naturally, the most intuitive one is that of words, though they are far from the only atomic unit of choice for this task. Indeed, this is a non-trivial task because of many factors. For instance, vocabularies (i.e., the set of unique words in a corpus of documents) might be too large and sparse when segmenting for words. Moreover, some languages do not have explicit word boundary markers (such as spaces for English).

This is a vast and interesting topic of its own, and we point readers to the work by Mielke et al. [24] for further information on it. Very briefly, it is important to mention that recent approaches rely on *sub-word* tokenization approaches, which broadly operate on the assumption that common words should be kept in the vocabulary, while rarer words should be split into "sub-word" tokens. This allows for smaller and more dense vocabularies and has been successfully applied to many recent approaches. Examples of sub-word tokenization approaches include Byte-pair Encoding [25], WordPiece [26], and SentencePiece [27]. Even more recently, some authors have argued for different forms of decomposition, such as ones utilizing underlying bytes [28], or even visual modeling based on the graphical representation of text [29].

### 2.1.2. Weighted Word Counts

Some of the more traditional and widely utilized approaches in the past are based on word occurrence statistics, effectively ignoring sentence structure and word semantics entirely. The most common example is that of the *Bag-Of-Words* (BoW) representation; in it, documents are simply represented as a count of their composing words, which are then often weighted with normalizing terms such as the well-studied Term Frequency-Inverse Document Frequency (TF-IDF) [21]. Briefly, TF-IDF measures word relevancy by weighting it positively by how frequently the word appears in a document (TF), but also negatively by how frequently it appears in all other documents (IDF). This way, words that frequently appear in a document are highlighted, but only if they do not appear often in other documents as well (as this negates their discriminative power).

### 2.1.3. Word Embeddings

Weighted word counts such as TF-IDF weighted BoW fail to capture any type of semantic and syntactic property of text. A major milestone towards the development of more meaningful representations is the development of *word embeddings*, initially popularized by works such as Word2Vec [30,31] and GloVe [22]. These vectorial representations of text are learned through unsupervised language modeling tasks; briefly, language modeling refers to the creation of statistical models created through word prediction tasks and has been studied for many decades [32]. The resulting *language models* (LMs) are useful in a variety of tasks, one of which is indeed the extraction of meaningful word and sentence representations.

The LMs utilized to develop word embeddings are based on shallow neural networks pre-trained on massive corpora of documents, allowing them to develop meaningful vectorial representations for words. The underlying semantic properties of these representations have often been exemplified through vector arithmetic operations, such as the classic example of $\vec{king} - \vec{man} + \vec{woman} \approx \vec{queen}$. Nonetheless, at a practical level, these vectors—i.e., word embeddings—can then be utilized as input features for downstream algorithms (e.g., classifiers), leading to vast improvements in terms of performance.

### 2.1.4. Contextualized Language Models

Word embeddings have sometimes been defined as "static", as their early iterations produced representations that were unable to disambiguate the meaning of a word with multiple meanings (i.e., polysemous words) [1]. An embedding for such a word, then, would be an average of its multiple meanings, leading to an inevitable loss of information. While much research has been dedicated to the addition of context to word embeddings (with notable results such as ELMO [33]), the introduction of the Transformer architecture [23] (see Section 2.2.3) has been certainly the most pivotal moment towards the development of contextualized LMs.

In layman's terms, these models contextualize word embeddings by studying their surrounding words in a sentence (the "context") [34–38]. A notable change of Transformer-based LMs over previous approaches is the lack of recurrence in their architectures, which instead utilize the attention mechanism [39,40] as their main component. This was a drastic change when considering that recurrent neural networks (RNNs) [41,42] were the go-to approach for text interpretation before the introduction of purely attention-based models, as they are particularly effective when dealing with sequential data. However, Transformers are much more parallelizable and also have been shown to scale positively with increased network depth, something that is not true for RNNs [43]. Therefore, larger and larger Transformer-based LMs can be built (both in terms of training data and model parameters), a practice that has seen widespread use in the most recent literature. Many recent models are now referred to with the moniker of *large language models* (LLMs) because of the massive amount of parameters they contain. Examples of this trend include GPT-3 (175 billion parameters) [38], LLaMA (65 billion parameters) [44], GShard (600 billion parameters) [45], and Switch-C (1.6 trillion parameters) [46]. More recently, some LLMs are being called *foundation LMs* because of the broad range of few-shot capabilities that allow them to be adapted to several tasks [47]. Within this context, researchers have been active in proposing LLMs for HTC tasks [48].

### 2.1.5. Classification

In terms of classification, traditional word representation methods such as TF-IDF (and, to a lesser extent, word embeddings) have been widely utilized as input features for traditional classification methods such as decision trees [49,50], support vector machines [51,52], and probabilistic graphical models (e.g., naive Bayes and hidden Markov models) [53,54]. The same can be said about word embeddings, which have, however, seen much greater use with specialized neural network architectures such as convolutional neural networks (CNNs) [55–58] and RNNs [59–61]. Transformer-based LMs such as

the Bidirectional Encoder Representations from Transformers (BERT) [34] and Generative Pre-trained Transformer (GPT) [36], on the other hand, have showcased outstanding classification results by passing the contextualized embeddings through a simple feed-forward layer. The ease of adaptation of these models has highlighted the importance of developing well-crafted representations for text. For a more in-depth description of classification approaches in NLP, we refer the readers to Gasparetto et al. [1].

### 2.2. Notable Neural Architectures

Much of the progress achieved in the development of meaningful text representation is attributable to neural networks, and deep learning in particular. While earlier approaches discussed (such as Word2Vec and GloVe) were initially based on shallow multilayer perceptrons [22,30], better results were later obtained with larger and deeper networks (i.e., more layers, more parameters). In this section, we briefly overview some of the most influential neural architectures and mention notable applications in the field of text representation.

#### 2.2.1. Recurrent Neural Networks

RNNs [41] are networks particularly well-suited to environments that utilize sequential data, such as text or time series. This particular architecture allows these networks to retain a certain amount of "memory", making them able to extract latent relationships between elements within sequences. In practice, this is performed by utilizing information from prior inputs to influence the current input and output of the network.

Simple RNNs for text processing are fed a sequence of word embeddings, which are processed sequentially. At each time step, the network receives both the next word vector as well as the hidden state of the previous time step (Figure 1). Unfortunately, because of their structure, standard RNN architectures are vulnerable to gradient-related issues, such as vanishing and exploding gradients [62]. In response to this issue, these architectures are frequently enhanced with *gating* mechanisms, the most popular being long short-term memory (LSTM) [63] and gated recurrent unit (GRU) [64]. Briefly, these gates allow the network to control which and how much information to retain, such as to enable the modeling of long-term dependencies. RNNs that utilize these gates are often referred to by the acronym of the gate itself, i.e., LSTM networks and GRU networks.

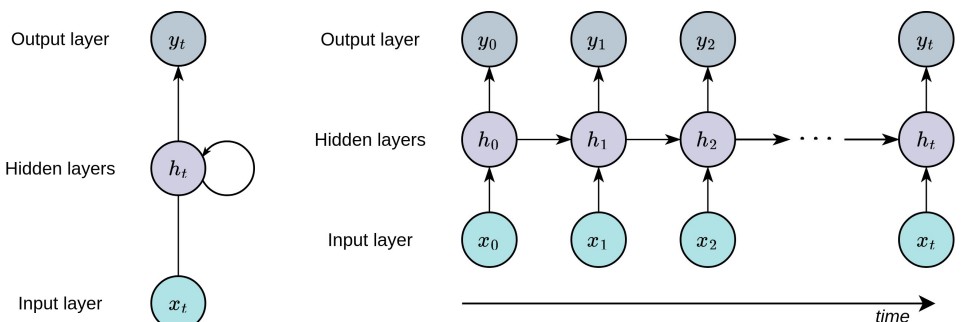

(**a**) Compressed basic RNN.      (**b**) Basic RNN shown unfolded at different time steps.

**Figure 1.** Exemplification of a simple RNN structure.

A simple RNN [65] can be defined as in the equation below, where $h_t$ and $x_t$ are the hidden state and input vector at time $t$, respectively, while $\boldsymbol{W_h}$ and $\boldsymbol{W_x}$ are learnable weight matrices:

$$h_t = \sigma(\boldsymbol{W_h}h_{t-1} + \boldsymbol{W_x}x_t + \boldsymbol{b_h}) \tag{1}$$

Here, $\sigma$ is an activation function, typically tanh or Rectified Linear Unit (ReLU), and $\boldsymbol{b}$ is a bias term. The output $y_t$ at each time step is usually derived from the state of the last layer of the RNN at all time steps, for instance:

$$y_t = \sigma(\boldsymbol{W_y}h_t + \boldsymbol{b_y}) \tag{2}$$

In the context of text representation, RNNs based on encoder–decoder architectures have been widely utilized to extract meaningful textual representations [66]. Briefly, an encoder–decoder structure can be understood as an architecture by which inputs are mapped to a compressed yet meaningful representation (contained in the hidden states between encoder and decoder). This representation should hopefully capture the most relevant features of the input and can then be decoded for a variety of different tasks (e.g., translation). Autoencoders [67] are a particular class of encoder–decoder network that attempts to regenerate the input exactly; they are particularly useful in creating efficient representations of unlabeled data.

In this context, the hidden states between the encoder and the decoder make for a semantically meaningful and compact representation of input words. The introduction of bidirectionality (influence in both the left-to-right and right-to-left directions) in RNNs has also been proved to be beneficial and has been used to achieve notable results, such as the aforementioned ELMo [33], a bidirectional LSTM-based language model that marked one of the first milestones towards the development of contextualized word embeddings. Nonetheless, RNNs have inherent limitations because of how they process data sequentially, making them ill-suited for parallelization. Furthermore, despite the improvements introduced by LSTM and GRUs, RNNs still struggle with long sequences because of memory constraints and their tendency to forget earlier parts of the sequence [62].

### 2.2.2. Convolutional Neural Networks

CNNs [68] are well-known neural architectures originally devised for computer vision applications. However, these networks have since been extended to other fields, achieving excellent results in NLP tasks as well [55,58]. The core structural element of CNNs is the *convolutional layer*, which applies a feature detector (*kernel* or *filter*) across subsets of the input (i.e., the convolution operation) to extract features. While this has a more intuitive interpretation in computer vision (where the filter moves across the image to search for features), the same reasoning can be applied to text. Intuitively, convolution as applied to images can be thought of as a weighted average of each pixel based on its neighborhood; the general idea of the process is outlined in Figure 2a. If we consider a vectorial representation of text (i.e., word embeddings), applying a filter as wide as the embedding size (a common approach) allows us to search for features within the sentence, as shown in Figure 2b. CNNs often use pooling operators (such as max or average) to reduce the size of the learned feature maps, as well as to summarize information.

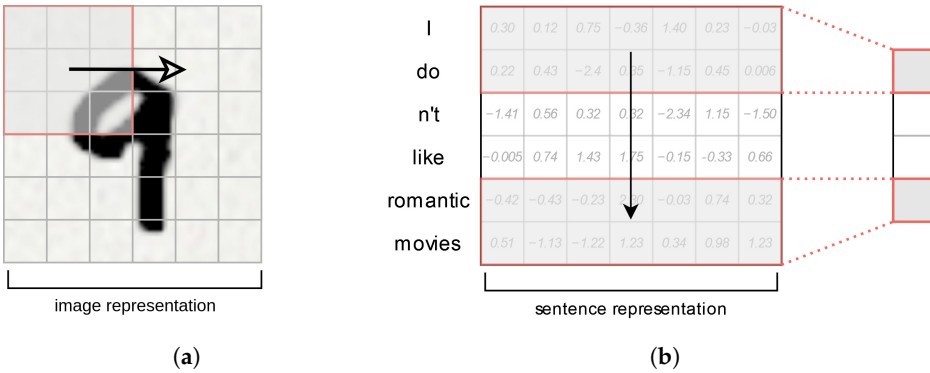

|   |   |
|:-:|:-:|
| image representation | sentence representation |
| (**a**) | (**b**) |

**Figure 2.** Exemplification of the application of convolutional filters in images and text. (**a**) A convolutional filter (top left) sliding across a digit from the MNIST dataset [69]. (**b**) A convolutional filter sliding across the vectorial representation of a sentence. Here, the filter is sliding in the conventional reading direction.

Various works have tested the efficacy of CNNs, especially as feature extractors on word embeddings [58]. An obvious upside of these architectures is their speed (as they are much more parallelizable than RNNs). Thus, CNNs produce efficient yet effective latent representations that can be used to solve a variety of tasks (e.g., classification). Recent

works have also revitalized the interest of CNNs in NLP by introducing temporal CNNs. In short, these aim to extend CNNs by allowing them to capture high-level temporal information [70,71].

2.2.3. Transformer Networks and the Attention Mechanism

As previously mentioned, one of the most influential neural architectures introduced in recent years—especially in terms of text processing—is the Transformer architecture [23]. The foundational framework is that of an encoder–decoder with a variable number of encoder and decoder blocks. In Section 2.1.4, we briefly touched upon the main innovation introduced by Transformers, which is the complete lack of recurrence as a learning mechanism. Instead, Transformers model context dependency between words entirely through the *attention mechanism* [39,40], which is outlined in the remaining part of this section.

Attention is, in essence, a weighting strategy devised to learn how different components contribute to a result. It was initially proposed in the machine translation domain [39] as an alignment mechanism that matched each word of the output (translated sentence) to the respective words in the input sequence (original sentence). The rationale behind this is that, when translating a sentence, a good translation can only be obtained by looking at the context of words and paying attention to specific words.

Vaswani et al. [23] used this mechanism in the Transformer architecture to allow the model to process all input tokens simultaneously, rather than sequentially, as was the case in previous recurrent networks. Input sequences are fed to the Transformer encoder at the same time, and a *positional encoding* scheme is used in the first layers of the encoder and decoder to inject some ordering information in the word embeddings. This ensures word ordering properties are not lost, e.g., that two words appearing in different positions in a sentence will have a different representation. Then, in all the remaining layers, Transformers use self-attention layers to learn dependencies between tokens. The layers are "self"-attentive, as each token pays attention to every other token in a sentence, which is the main learning mechanism that allows for the disposal of recurrence. Indeed, since tokens in the input sequence are being processed simultaneously, the encoders can look at surrounding tokens in the same sentence and produce context-dependent token representations [1].

Stacking several self-attention layers produces a multi-head attention (MHA) layer, whose structure is shown in Figure 3. Vaswani et al. [23] argue that having multiple attention heads in the layer enables the model to pay attention to different information in distinct feature spaces and at different positions. Input sequences in the attention heads are transformed using linear transformations to generate three different representations, which the authors name $Q$ (queries), $K$ (keys), and $V$ (values), following the naming convention used in information retrieval (as in Equation (3)):

$$Q = XW_Q, \qquad K = XW_K, \qquad V = W_V \tag{3}$$

where $W_Q, W_K \in \mathbb{R}^{dim \times d_k}, W_V \in \mathbb{R}^{dim \times d_v}$ are the learned weight matrices. In the Transformer architecture, $K$ and $V$ are always generated from the same sequence (as we will discuss, $Q$ is used differently in the encoder and decoder part). The transformed sequences are then used to compute the scaled dot-product attention, as follows:

$$Z_k = \text{ScaledDotAttention}(Q, K, V) = \text{softmax}\left(\frac{QK^T}{\sqrt{d_k}}\right)V \tag{4}$$

While many definitions of attention exist in the literature, the authors decided to use the scaled dot-product, mainly for efficiency reasons. The product of the key and query matrices is scaled by $\sqrt{d_k}$ to improve the stability of the gradient computation. At an intuitive level, we may envision a word $q \in Q$ as being queried for similarity/relatedness against keys $k \in K$, finally obtaining the relevant word representation by multiplying by $V$.

The final result of the MHA layer is the concatenation of all heads multiplied by matrix $W_O \in \mathbb{R}^{hd_v \times dim}$, which reduces the output to the desired dimension $\mathbb{R}^{N \times dim}$:

$$Z = \text{Concat}(Z_k)W_O \tag{5}$$

In encoder blocks, $Q$, $K$, and $V$ are all generated by the input sequence and have the same size; the general structure of an encoder block can be seen in Figure 4a. In decoder blocks, $Q$ comes from the previous decoder layer, while $K$ and $V$ come from the output of the associated encoder. These blocks structurally differ from encoders in the presence of another MHA layer, which precedes the standard one and is introduced to mask future tokens in a sentence (Figure 4b). This allows the decoder to be "autoregressive" during training, as it otherwise would trivially look forward in a sentence to obtain the result.

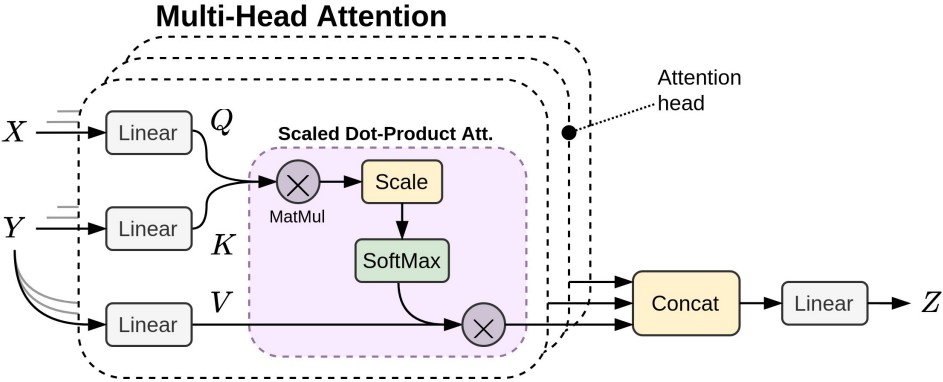

**Figure 3.** The multi-head attention layer used in the Transformer architecture.

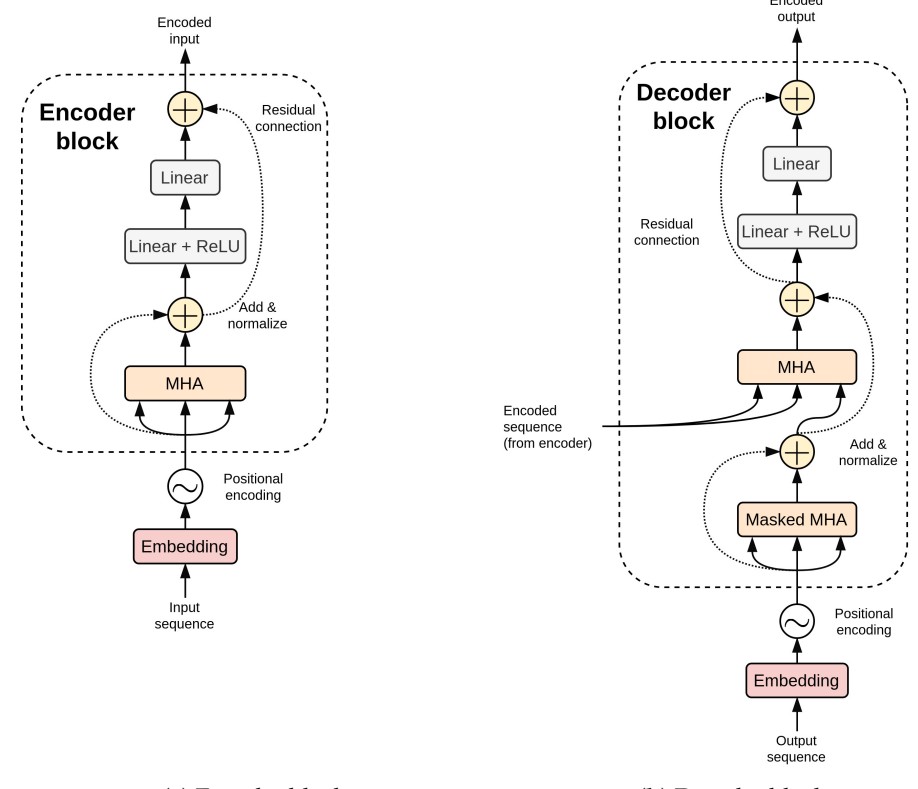

(**a**) Encoder block.　　　　(**b**) Decoder block.

**Figure 4.** Transformer encoder–decoder architecture.

With the widespread usage of Transformer blocks with MHA, researchers have studied the features extracted at each layer. Interestingly, some of these works found that each encoding block may focus on the extraction of linguistic features at different levels: syntactic features are mostly extracted in the first blocks, while deeper layers progressively focus on semantic features [72,73]. This "layer specialization" phenomenon suggests that stacking attention layers creates expressive representations that blend morphological and grammatical features.

The attention mechanism served as the basis of various derived attention schemes. For example, in the TC (including HTC) domain, hierarchical attention networks (HANs), proposed by Yang et al. [74], have been widely used [75–77]. Their idea is to apply attention hierarchically at different levels of granularity. In many applications, the mechanism is initially applied to words and used to produce more informative sentence representations, and then it is also applied to sentences to produce better document representations. This method adapts particularly well to the processing of long texts, since it leverages the hierarchical nature of human-produced written documents, which are often structured hierarchically.

### 2.2.4. Graph Neural Networks

The ubiquity of graph structures in most domains has sparked much interest in the application of neural networks directly to graph representations. In the NLP domain, a body of text can be represented as a graph of words, where connections represent relations that are potentially semantic or grammatical. As a simple example, a sentence could be represented by word nodes, and adjacent words in the sentence would be linked by an edge. Entire documents can also be linked together in a graph, for instance, in citation networks, or simply considering their relatedness [1,78].

In the particular context of hierarchical classification, graphs of labels have often been used to propagate hierarchy information between connected labels, with connections usually representing parent–child relations.

### Message Passing

The principle behind graph processing models is generalized by the message passing neural network [79]. In this model, a *message passing phase,* lasting $T$ time steps, is used to update node and edge representations by propagating information along edges. First, for a graph $G = (V, E)$, the message at time $t + 1$ is computed for each node $u \in V$, depending on the previous values of the nodes and edges $e \in E$. Therefore, for a node $u$,

$$m_u^{t+1} = \sum_{v \in \mathcal{N}(u)} \phi(x_u^t, x_v^t, e_{uv}^t) \tag{6}$$

where $\phi$ is a function learned by a neural network, function $\mathcal{N}(u)$ gives the neighbor nodes of $u$, and $x \in X$ are node embeddings (representations). Equation (6) uses the sum operation to aggregate the multiple messages coming from the neighbors of $u$, although it could be replaced with other permutation-invariant operations, such as the average or minimum. Once the messages have been computed, the node embeddings are updated using an update function $\sigma$, which is also learned by a neural network:

$$x_u^{t+1} = \sigma(x_u^t, m_u^{t+1}) \tag{7}$$

Analogously, Equations (6) and (7) could be adapted to compute the message starting from neighbor edges instead of nodes and to update the edge representation accordingly. Finally, in the *readout phase*, values from all nodes are aggregated by summing them together and the output is used for a graph-level classification task.

### Graph Convolution

The concept of message passing lends itself to the definition of a convolution operation for graphs that can be used within graph convolutional neural networks (GCNs) [80]. In

the literature, two main categories of GCNs are typically distinguished: spatial-based and spectral-based. Similarly to the conventional convolution operation over an image, spatial-based GCNs define graph convolutions based on the graph topology, while spectral-based methods are based on the graph's spectral representation [78,81,82]. In this paragraph, we will only discuss the former type, which is more closely related to the message-passing concept.

While many different definitions of convolution have been proposed, a simple spatial convolution operator on a graph can be defined as:

$$\text{Conv}(x_u^t) = \sigma\left(\bigodot_{v \in \mathcal{N}(u)} \phi(x_u^t, x_v^t)\right) \tag{8}$$

in which $\odot$ is a permutation invariant operation. The result of this operation can be used to update each node representation, as exemplified in Figure 5. When $\odot = \sum$ and $\phi(x_u^t, x_v^t) = x_v^t \mathbf{W}^T + b$, which is a simple linear transformation of the representation of neighbor nodes, this operation can be defined in matrix form as:

$$\text{Conv}(\boldsymbol{X}) = \sigma(\boldsymbol{A}(\boldsymbol{X}\boldsymbol{W}^T + \boldsymbol{b})) \tag{9}$$

In the equation above, $\boldsymbol{A} \in \mathbb{R}^{n \times n}$ is the adjacency matrix, $\boldsymbol{X} \in \mathbb{R}^{n \times d}$ contain the nodes' representation of dimensionality $d$, and $\boldsymbol{W} \in \mathbb{R}^{d_{out} \times d}$ and $b \in \mathbb{R}^{d_{out}}$ are the weight matrix and bias term, respectively. The multiplication by the adjacency matrix $\boldsymbol{A}$ guarantees that only neighbor nodes contribute to the updated representation $\boldsymbol{X}$. Consequently, if multiple convolutions are stacked, the message from each node can propagate further in the graph. However, if too many layers are used, large portions of the graph could end up having similar node representations, an issue often referred to as *oversmoothing* [83].

As a natural extension of the operation defined in Equation (9), $\boldsymbol{A}$ can be weighted to reflect the importance of neighbors, for instance, by multiplying each entry with edge weights. The attention mechanism can also be used to autonomously learn how much each node should contribute to their neighbors' representations. An example of this is graph attention networks (GATs) [84], which use the Transformer's MHA to compute the hidden states of each node.

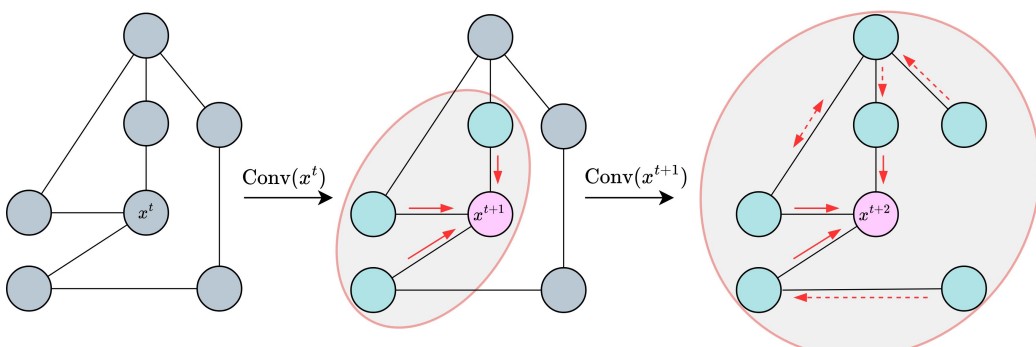

**Figure 5.** The propagation effect operated by two sequential graph convolutions concerning node $x$. The red arrows showcase the information flow toward the target node.

### 2.2.5. Capsule Networks

Much recent literature explores the usages of capsule networks (CNs) [85] to find structure in complex feature spaces. These networks group perceptrons—the base units of feed-forward networks—into *capsules*, which can essentially be interpreted as groups of standard neurons. Each capsule specializes in the identification of a specific type of entity, like an object part, or, in general, a concept. Since a capsule is a group of neurons, its output

is a vector instead of a scalar, and its length represents the probability that an entity exists in the given input, as well as its spatial features.

Capsules have been first applied to object recognition [86] to address the shortcomings of CNNs, such as the lack of rotational invariance. However, capsules have been proposed in NLP applications as well; for instance, TC datasets often present labels that can be grouped into related meta-classes that share common concepts. A hierarchy of topics may include a "Sports" macro-topic, with "Swimming", "Football", and "Rugby" as sub-topics. However, the latter two sports are more similar, both being "team sports" and "ball sports", something that is not made explicit by the hierarchy. CNs have been applied to HTC tasks for their ability to model latent concepts and, hence, capture the latent structure present in the label space. It is expected that a better understanding of the labels' organization, such as modeling the "team sports" and "ball sports" concepts, can be effectively exploited to improve decision-making during classification [87,88].

### Capsules

As briefly mentioned above, a capsule is a unit specialized in the following tasks:

- Recognizing the presence of a single entity (e.g., deciding how likely it is that an object, or piece-of, is present in an image);
- Computing a vector that describes the instantiation parameters of the entity (e.g., the spatial orientation of an object in an image or the local word ordering and semantics of a piece of text [88]).

As a result, capsules can specialize and estimate parameters about the selected entities. In contrast, neurons in standard MLPs can only output scalar values that cannot encapsulate such a richness of information. Within CNs, capsules are organized in layers, and the output of each child capsule is fed to all capsules in the next layer (i.e., parent capsules) using weighted connections that depend on the level of agreement between child capsules. Higher layers tend to specialize in recognizing more high-level entities, estimating their probability using the information about sub-parts that are propagated by lower-level capsules [89]. Output between capsules is routed through the *dynamic routing* mechanism.

### Dynamic Routing

Because each layer of capsules specializes in recognizing specific entities, subsequent layers should make sense of the information extracted by previous layers and use it to recognize more complex entities. However, not all entities detected in lower-level capsules may be relevant. Hence, a *routing-by-agreement* method was proposed to regulate the flow of information to higher layers.

The general principle behind the routing-by-agreement algorithm known as dynamic routing [89] is that connections between capsules in layer $l$ and a capsule $j$ in layer $l + 1$ should be weighted depending on how much the capsules in $l$ collectively agree on the output of capsule $j$. When many of them agree, it means that all the entities they recognized can be part of the composite entity recognized by capsule $j$, and as such, their output should be sent mostly to capsule $j$.

The routing algorithm updates the routing weights (i.e., connections between capsules in different layers) so that they reflect the agreement between them. Let $\{\mathcal{K}, \mathcal{L}\}$ be two subsequent layers of a CN, each made up of several capsules. Predictions made by capsule $i \in \mathcal{K}$ about the output of capsule $j \in \mathcal{L}$ are computed as:

$$\hat{\boldsymbol{u}}_{j|i} = \boldsymbol{W}_{ij}\boldsymbol{u}_i \tag{10}$$

where $\boldsymbol{u}_i$ is the activation vector of capsule $i$ and matrix $\boldsymbol{W}$ is the learned transformation matrix that encodes the part-to-whole relationship between pairs of capsules in two subsequent layers. A distinct weight matrix $\boldsymbol{B}$ is used to store the connection weights $\boldsymbol{B}_{ij}$ between

each capsule $i \in \mathcal{K}$ and capsule $j \in \mathcal{L}$. The agreement is measured using the dot product, and weights are updated as in the equation below:

$$B_{ij} \leftarrow B_{ij} + \hat{u}_{j|i} v_j \tag{11}$$

where output $v_j$ represents the probability that the entity recognized by capsule $j$ is present in the current input and therefore, should be consistent with the information extracted by lower-level capsules (i.e., the "guesses"), which is encoded in vector $\hat{u}$. Hence, the output of a capsule $j \in \mathcal{L}$ is the weighted sum of the predictions made by all capsules in the previous layer $\mathcal{K}$. Note that, since the activation vector must represent a probability, the *squash* function is used to shrink the output into the $(0, 1)$ range:

$$v_j \leftarrow \text{squash} \left( s_j \right) \qquad s_j = \sum_{i \in \mathcal{K}} C_{ij} \hat{u}_{j|i} \tag{12}$$

During each iteration, the *coupling coefficients* (or *routing weights*) are computed as follows:

$$C_i \leftarrow \text{softmax} \left( B_i \right), \qquad \forall \text{ capsule } i \in \mathcal{K} \tag{13}$$

This process ensures that the capsules in $\mathcal{K}$ that were more in agreement with the capsules in $\mathcal{L}$ will send a stronger signal than the capsules that made a different prediction, as opposed to higher-level capsules. Moreover, as Equation (12) shows, the output of the capsules in subsequent layers is dependent on the prediction made in the previous layers. This means that the higher the number of capsules that agree on the most likely prediction of some capsules in the next layer (i.e., $\hat{u}$), the more they can influence the output of those capsules.

The routing process is repeated a fixed number of times before the algorithm proceeds to the next layer and stops when all the connections between capsules are weighted. Finally, this routing mechanism has been recently improved using the expectation-maximization algorithm [90] to overcome some of the limitations of the former approach.

## 3. Hierarchical Text Classification

As a sub-task of the broader TC area, HTC methods have much in common with standard text classifiers. In this section, we will outline the main aspects that differentiate HTC from standard classification approaches and how they can be leveraged to achieve a higher classification accuracy in the presence of a taxonomy of labels.

### 3.1. Types of Hierarchical Classification

Standard classifiers focus on what the HTC literature defines as *flat classification*. In it, categories are treated in isolation (i.e., as having no relationship between one another) [15,16]. In contrast, HTC deals with documents whose labels are organized in structures that resemble a tree or a directed acyclic graph (DAG) [91,92]. In these structures, each node contains a label to be assigned, such as in Figure 6. Methods able to work with both trees and DAGs can be devised, though a simpler technique is to simply "unroll" or "flatten" sub-nodes with multiple parents for DAGs, thus obtaining a tree-like representation. For this reason, this article (and much of the HTC literature) focuses on hierarchies with a tree structure.

HTC approaches can be divided into two groups: *local* and *global* approaches [93]. Local approaches (sometimes called "top-down") are defined as such because they "dissect" the hierarchy, constructing multiple local classifiers that work with a subset of the node labels. While more informed than flat classifiers—which ignore the hierarchy—there is an inevitable loss of hierarchical information, as the aggregation of these classifiers tends to ignore the holistic structural information of the taxonomic hierarchy. Depending on the chosen approach, the amount of information regarding the hierarchy can be partial or absent [94]. Local approaches (Figure 7b–d,) have been criticized because of their structural issues, the most notable one being that they may easily propagate misclassifications [95,96].

Furthermore, these models are often large in terms of trainable parameters and may easily develop exposure bias because of the lack of holistic structural information [97]. While we will discuss this in more detail later, this arises from the fact that, at test time, lower-level classifiers use the prediction of previous classifiers, thus leading to a discrepancy in the training process (which is usually based on ground truths).

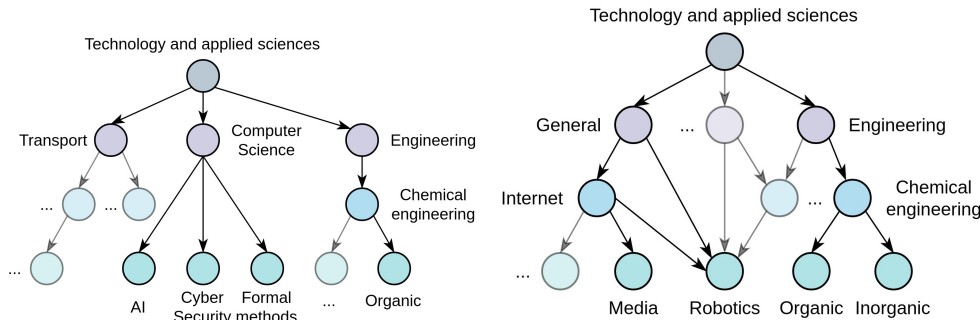

(**a**) Label hierarchy based on a tree structure.  (**b**) Label hierarchy based on a DAG structure.

**Figure 6.** Hierarchically structured labels (inspired by Wang et al. [91]). Both forms are frequent in practice, though labels organized in a DAG require adaptation of either the method or the structure.

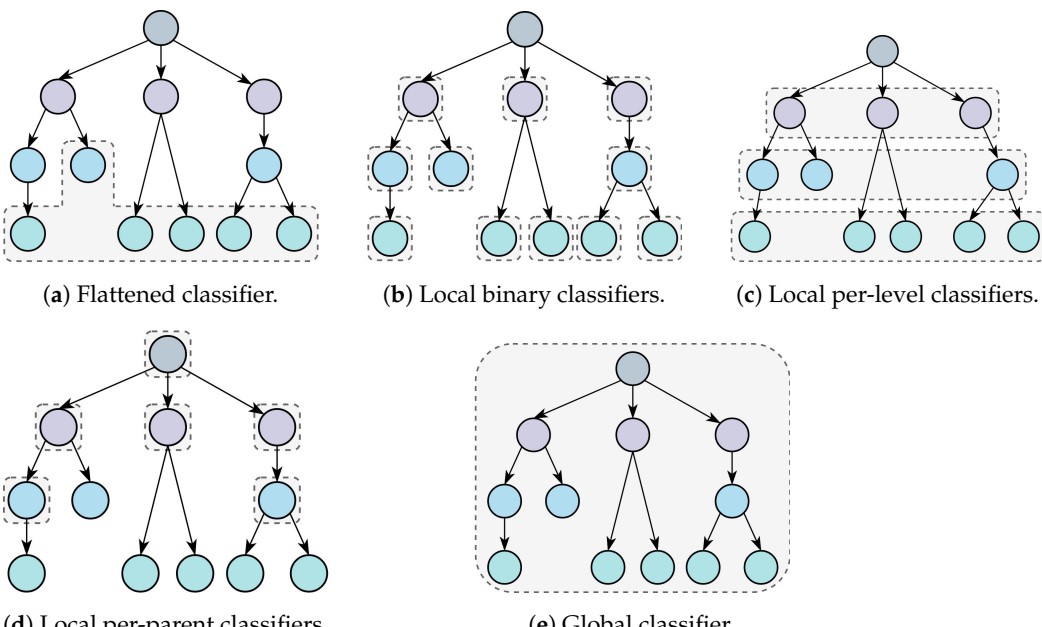

(**a**) Flattened classifier.   (**b**) Local binary classifiers.   (**c**) Local per-level classifiers.

(**d**) Local per-parent classifiers.          (**e**) Global classifier.

**Figure 7.** The most common approaches to HTC, exemplified on a tree-like hierarchy. Flattened classifiers (**a**) lose all hierarchical information, while local classifiers (**b**–**d**) can incorporate some of this information. Global classifiers (**e**) aim to fully exploit the label structure.

Global methods aim to solve these shortcomings, as well as being frequently less computationally expensive (as there is a single classifier) [17]. While the definition of global classifiers is deliberately generic, one might imagine as global any individual classification algorithm that directly takes into account the hierarchy. It is also worth noting that global and local approaches can also be combined [98,99]. Figure 7 showcases the main approaches to HTC, which we now briefly summarize.

### 3.1.1. Flattened Classifiers

The flat classification approach (Figure 7a) reduces the task to a multiclass (or multilabel) classification problem, therefore discarding hierarchical information entirely. Typically,

only leaf nodes are considered, and the classification of higher levels of the hierarchy is inherited from parent nodes [88,94].

### 3.1.2. Local Classifiers

Local classification approaches are generally divided into three categories that differ in how they dissect the hierarchy. First, the class of *local per-node (binary)* approaches (Figure 7b) considers each label as a separate class, disregarding the hierarchy completely. This approach is among the simplest and resembles a one-versus-rest approach [14]. The individual classifiers receive hierarchical information from the specific training and testing phases, which we outline in Section 3.1.4. *Local per-level* methods (Figure 7d) assign a classifier to each relevant category level, which is tasked to decide for that level alone. Finally, *local per-parent* methods (Figure 7d) assign a classifier to all parent nodes, tasking them to assign a sample to one of its children—therefore capturing part of the sample's current path through the hierarchy.

### 3.1.3. Global Classifiers

A global (or big-bang) [19] classification approach (Figure 7e) makes use of the entire hierarchy to make the final classification decision. It is common (though not strictly necessary) for global classifiers to perform actual classification on a flattened representation; hierarchy information is therefore achieved through structural bias (i.e., the architecture of the model) [17].

### 3.1.4. Training and Testing Local Classifiers

Differently from standard (flat) classification approaches, local classifiers usually require specialized procedures for training and testing, both in terms of actual methods and in the definition of positive and negative examples, as outlined in the following paragraphs.

#### Testing

Testing phases are usually characterized by a "top-down" flow of information. This is the preferred approach to all local approaches. The system performs a prediction at the first, most generic level, then forwards the decision to the children of the predicted class. As mentioned, this makes these approaches vulnerable to error propagation (where a mistake at higher levels leads to an inevitable mistake at the lower ones), unless a specialized procedure is set in place to avoid this.

Local classifiers can be used independently, therefore lending themselves naturally to multilabel scenarios. However, this might lead to class-membership inconsistency (by which a child node disagrees with a parent classification); therefore, top-down approaches are usually utilized.

#### Training

As local classifiers are defined at different levels of the hierarchy, examples may need to be altered so that the objective makes sense at the local level. Several possible policies may be utilized for the creation of these subsets of examples, each of which differs in how "inclusive" they are. For instance, a decision can be made on whether or not to include samples labeled with "AI" as positive examples for classifiers at the higher "Computer Science" level in Figure 6a (while everything else is considered as a negative example). We point interested readers to Silla and Freitas [17] for an in-depth overview of these policies.

It is also possible to allow for different classification algorithms in different nodes, an approach that is often attributed to per-parent approaches [17]. To do this, training data may be further split into sub-train and validation sets, and the best decider for each node is selected dynamically.

### 3.2. Non-Mandatory Leaf Node Prediction and Blocking

In hierarchical classification datasets, it may not always be the case that all prediction targets correspond to leaves. Many authors distinguish between *mandatory* and *non-mandatory leaf node prediction* (MLNP, NMLNP) [15,100]. Quite simply, in NMLNP scenarios, the classification method should be able to consider stopping the classification at any level of the hierarchy, regardless of whether it is a leaf node or not. The term applies to both tree- and DAG-structured hierarchies. In this section, we briefly outline how the different types of HTC approaches can deal with the latter, more complex case.

#### 3.2.1. Flattened Classifiers

The flat classification approach simplifies the problem to a standard, non-hierarchical classification problem. In that sense, if the target is restricted to the leaf nodes of the structure, methods that follow this paradigm are unable to deal with NMLNP by design [17]. It is possible to naively extend the classification targets to include all possible labels, though this would make the task much harder since flattened classifiers do not have any inherent information about the hierarchy. As we will discuss in the following paragraphs, it is possible to inject hierarchical information whilst maintaining the general classification target and algorithm, which some global approaches do [101].

#### 3.2.2. Local Classifiers

To deal with NMLNP, local approaches must implement a *blocking mechanism*, so that inference may be stopped at any level of the hierarchy. A simple way to do this is by utilizing a threshold on the confidence of each classifier; if during top-down prediction the confidence does not meet the requirement, the inference process is halted [102].

An issue with such thresholds is that they may lead to incorrect early stopping of classification. Sun et al. [103] define this as the *blocking problem*, which refers to any case in which a low confidence rating of a parent classifier mistakenly hints that an example does not belong to its actual macro-class. As a consequence, the example will never reach the classifier for its appropriate sub-class. The authors propose some blocking reduction methods, which generally act by reducing the thresholds of inner classifiers or by allowing lower-level classifiers to have a second look at rejected examples.

#### 3.2.3. Global Classifiers

Global classifiers utilize a single, usually complex algorithm that integrates the hierarchy into its internal reasoning. As mentioned, it is common to base global classifiers on an existing flat classification approach and modify it to take into account the class hierarchy. Global classifiers can also be used in NMLNP scenarios, though specific strategies might be needed in this case to enforce class-membership consistency in predictions. Likewise, it is also possible to integrate a top-down prediction approach (which would be internal to the algorithm) to avoid this issue.

### 3.3. Evaluation Measures

As HTC issues are inherently multiclass (or multilabel), many researchers choose to utilize standard evaluation metrics widely adopted in classification scenarios. As mentioned, however, many authors argue that these measures are inappropriate [15,17,18,104–106]. Intuitively, these arguments are based on the shared belief that ignoring the hierarchical structure in the evaluation of a model is wrong because of the concept of *mistake severity*; in other words, a model that performs "better mistakes" should be preferred [107]. This follows from two considerations. Firstly, predicting a label that is structurally close to the ground truth should be less penalizing than predicting a distant one. Secondly, errors in the upper levels of the hierarchy are inherently worse (e.g., misclassifying "football" as "rugby" is comparatively better than misclassifying "sport" as "food"). These considerations also make sense when considering real-world applications of HTC, such as ICD coding [5,6] and legal document concept labeling [7]. A better mistake entails that most ancestor nodes

in the prediction path were correct, meaning that most of the macro categorizations of the sample were accurate. In most scenarios, a mistake of this type is preferable to one in which the macro category is wrong, the latter of which might have severe consequences. Moreover, understanding the severity and type of a model's errors can aid during the development process and possibly lead to new strategies to prevent them.

In this section, we will provide an outline of the most common evaluation metrics utilized in HTC, both standard and hierarchical. For a more in-depth analysis of these metrics, as well as of the issues they present and how to address them, we refer to Kosmopoulos et al. [105].

### 3.3.1. Standard Metrics

The most common performance measures utilized in "flat" classification are derived from the classic information retrieval notions of *accuracy* (*Acc*), *precision* (*Pr*), and *recall* (*Re*). As in any other supervised learning task, we consider the truthfulness of a model's predictions against the ground truth derived from the dataset. Formally, let $\{(x_0, y_0), \ldots, (x_n, y_n)\}$ be a set of labeled training examples, where $x \in \mathbb{R}$ is an input example and $\mathbf{y} \in \{0, 1\}^L$ is the associated label vector, with $L$ being the set of label indices (therefore, $|L|$ is the number of categories). For each label $l \in L$, we can calculate category-specific metrics by considering positive (P) and negative (N) predictions for each example. A prediction is considered true (T) if it agrees with the ground truth and false (F) otherwise.

*Accuracy* measures the ratio of correct predictions over the total of number predictions. *Precision* is instead a measure of correctness, quantifying the proportion of true positive predictions among the ones made, while *recall* is a measure of completeness, quantifying the proportion of overall positives captured by the model. Notably, the latter two metrics have a larger focus on the impact of false predictions. Accuracy, precision, and recall are defined as follows (Equation (14)):

$$Acc = \frac{|TP| + |TN|}{|TP| + |TN| + |FP| + |FN|} \qquad Pr = \frac{|TP|}{|TP| + |FP|} \qquad Re = \frac{|TP|}{|TP| + |FN|} \tag{14}$$

Precision and recall do not effectively measure classification performance in isolation [3]. Therefore, a combination of the two is commonly utilized. The F-measure ($F_\beta$) is the most popular of these combinations, providing a single score according to some user-defined importance of precision and recall (i.e., $\beta$). Normally, $\beta = 1$, resulting in the harmonic mean of the two measures (Equation (15)):

$$F_\beta = \frac{(\beta^2 + 1) \cdot Pr \cdot Re}{\beta^2 \cdot Pr + Re} \qquad F_1 = 2 \cdot \frac{Pr \cdot Re}{Pr + Re} \tag{15}$$

Accuracy naturally extends to multiclass settings without being able to weigh the contribution of class differently. Precision and recall can be averaged in different ways; *macro* averaging considers all class contributions equally (Equation (16)), while *micro* averaging treats all examples equally (Equation (17)):

$$Pr_{macro} = \frac{\sum_{i=1}^{m} Pr_i}{m} \qquad Re_{macro} = \frac{\sum_{i=1}^{m} Re_i}{m} \tag{16}$$

$$Pr_{micro} = \frac{\sum_{i=1}^{m} |TP_i|}{\sum_{i=1}^{m} |TP_i| + |FP_i|} \qquad Re_{micro} = \frac{\sum_{i=1}^{m} |TP_i|}{\sum_{i=1}^{m} |TP_i| + |FN_i|} \tag{17}$$

where $m$ is the number of categories. Micro-averaging may be useful when the class imbalance is severe and needs to be accounted for in the measurements. Support-weighted metrics are also an alternative in such cases. F-measures may be micro- or macro-averaged as well, utilizing the corresponding averaged versions of precision and recall.

### 3.3.2. Hierarchical Metrics

As outlined before, standard metrics lack the capability of reflecting the relationships that exist among classes. In this context, Sun and Lim [15] propose to solve this issue by introducing hierarchical metrics based on category *similarity* and *distance*. Category similarity evaluates the cosine distance of the feature vectors representing predicted and true categories, while category distance considers the number of links between the two in the hierarchy structure. While interesting, these measures have practical issues, which Kiritchenko et al. [104] outline (such as inapplicability to DAGs and multilabel tasks). Instead, they propose to extend traditional precision and recall. To do this, they augment the set of predicted and true labels to include all their ancestors (Figure 8) and calculate the metrics on the augmented sets.

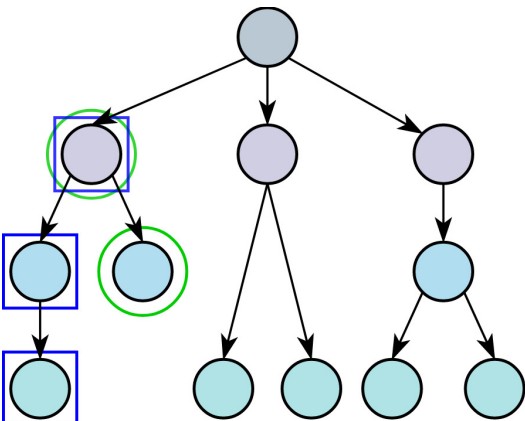

**Figure 8.** Tree hierarchy with predicted (squares) and ground truth (circles) labels, with the ancestors of each highlighted. As the two sets have a node in common, the misprediction should be considered less severe.

Formally, let $\hat{Y}_{aug}$ be the augmented set of predictions containing the most specific predicted classes and all of their ancestors and $Y_{aug}$ the augmented set of the most specific ground truth class(es) and all of their ancestors. Then, hierarchical metrics (h-metrics) can be defined as follows (Equation (18)):

$$hPr = \frac{\sum_i |\hat{Y}_{aug} \cap Y_{aug}|}{\sum_i |\hat{Y}_{aug}|} \qquad hRe = \frac{\sum_i |\hat{Y}_{aug} \cap Y_{aug}|}{\sum_i |Y_{aug}|} \qquad hF_\beta = \frac{(\beta^2 + 1) \cdot hPr \cdot hRe}{\beta^2 \cdot hPr + hRe} \qquad (18)$$

However, Kosmopoulos et al. [105] observe that this approach overpenalizes errors in which nodes have many ancestors. Specifically, false positives predicted at lower levels of the hierarchy strongly decrease the precision metric, while recall tends to be boosted, since adding ancestors is likely to increase the number of true positives (false positives are ignored). Drawing from the lowest common ancestor (LCA) concept defined in graph theory [108], they propose LCA-based measures as a solution. Briefly, and in the context of tree structures, the LCA of two nodes can be defined as the lowest (furthest from the root) node that is an ancestor of both. Therefore, these new measures are defined similarly to the one in Equation (18), but with the expanded sets defined in terms of LCA rather than on full node ancestries [18]. Despite these issues, the hierarchical metrics of Equation (18) are still considered effective measures across a broad range of scenarios [17].

Some authors report metrics that explicitly evaluate the difference, in terms of path distances through the hierarchy, between class predictions and ground truth labels. In particular, Sainte Fare Garnot and Landrieu [109] define the average hierarchical cost (AHC) of a set of predictions. Given two labels $a, b \in L$, define dist$(a, b)$ as the number of edges that separate two labels in the hierarchy tree. In turn, for a set of labels $S$, let dist$(a, S)$ be the minimum distance between the node and the set, i.e., $min(dist(a, s) \; \forall s \in S)$. If

predictions and ground truth labels are expressed as vectors $\in \{0, 1\}^L$ (as before), the AHC can be defined as (Equation (19)):

$$AHC(Y, \hat{Y}) = \frac{1}{|Y|} \sum_j \hat{\mathbf{y}}_j \, \text{dist}(j, Y) \tag{19}$$

In practice, this measure evaluates how "far off" the predictions were from the closest common ancestor (and, in that, are not too dissimilar for LCA-based measures). As a consequence, a low AHC indicates that mistakes were, on average, not too distant from the ground truth.

### 3.3.3. Other Metrics

Depending on the particular aspect of HMC being tackled, other types of metrics, often borrowed from neighboring disciplines, may be warranted. In this section, we briefly introduce them.

Some works report rank-based evaluation metrics, which are widely used in the more general context of multilabel classification [110,111]. Among many, the two most commonly utilized are precision and normalized discounted cumulative gain at *k* (*Pr@k*, *NDCG@k*). Briefly, given a model's prediction (i.e., a probability vector across labels), we can sort the labels in descending order based on the output probabilities. Then, *Pr@k* indicates the fraction of correct predictions in the top-*k* labels of this sorted list, whereas *NDCG@k* measures the ranking quality (i.e., how high correct labels are ranked). For a more precise description of these metrics, which are often utilized in recommendation systems and, more generally, in information retrieval, see Marcuzzo et al. [112].

In methods able to categorize labels at different levels of the hierarchy separately, many authors choose to showcase the accuracy score at each separate level, as well as a single overall score [87,113]. The overall score is the one obtained by classifying the last level of the hierarchy given the (possibly incorrect) predictions of the parent classes. In some cases, authors utilize *subset* accuracy, where all labels must match the ground truth exactly.

Lastly, some authors utilize the *Hamming loss* [114,115] metric for HTC. This measure evaluates the fraction of misclassified instances, i.e., true labels that were not predicted or predictions that were not in the ground truth. A comprehensive review of these metrics, as well as less-commonly utilized ones, may be found in Zhang and Zhou [116].

## 4. Hierarchical Text Classification Methods

In this section, we provide an overview of approaches devised to solve HTC tasks in the 2019–2023 period. Section 4.1 introduces the main class of methods and Section 4.2 lists all relevant works we found in our literature review. Finally, a subset of these is analyzed in more detail in Section 4.3.

### 4.1. Overview of Approaches

In this section, we provide a high-level overview of notable approaches for HTC that differentiate themselves from more standard approaches for TC and that we find to be popular within the literature. The categories presented are not hard partitions, but rather emergent categories with possible overlap.

*Sequence to sequence approaches*, as the name suggests, re-frame HTC as a Seq2Seq problem, where the input is the text to classify and the output is the sequence of labels, such as to obtain better modeling of the local label hierarchy [117]. Target labels for each text are flattened to a linear label sequence with a specific order via either sorting, depth-first search, or breadth-first search. Each prediction step generates the next label based on both the input text and the previously generated label; this way, the model can capture the level and path dependency information among labels. Several frameworks have been proposed utilizing this approach [118,119], many arguing for improved label generation and better use of co-occurrence information [120–122].

*Graph-based approaches* encode the hierarchy with specialized structure encoders, which allow for better utilization of the mutual information among labels. Tree-based and graph-based approaches have been proposed to encode the hierarchy, with both showcasing good results [97,123]. Several methods apply GCNs to the hierarchy of labels, often in a joint manner with the encoding of the text to enrich the representation [124]. Additional techniques are often applied to improve label correlation and to better model their co-occurrence [125–128]. Some authors, such as Ning et al. [129], argue that graph-based approaches may lose on the directed nature of the tree hierarchy and thus introduce unidirectional message-passing constraints to improve graph embedding. Other works utilize GCNs to reinforce feature sharing among labels of the same level [130]. While graph NNs and GCNs in particular are by far the most popular architecture in these approaches, some authors utilize Transformer-based approaches such as Graphormers [91]. All in all, graph-based approaches have obtained excellent results, and the integration of structural information in classification frameworks has proved very successful.

*KG-based approaches* argue that the integration of external knowledge can enhance classification. They attempt to incorporate this knowledge into the model using graph-based representations, known as knowledge graphs (KGs). A KG comprises nodes that represent real-world concepts or entities. The relationships between these concepts are depicted by edges, which can vary in type depending on the relationship they represent [112]. Well-established examples of KGs in the NLP community include DBpedia [131] and Concept-Net [132]. These methods, while technically a subset of the earlier mentioned graph-based approaches, distinguish themselves by utilizing KGs to encode external knowledge, as opposed to problem-specific data. KG-based methodologies typically utilize KGs to generate knowledge-aware embeddings. Several techniques have been proposed to encode entity–relation–entity triples [133,134]. These embeddings encapsulate the relationships between entities and effectively augment the representation with data that would not be accessible in a task-specific setting, thereby enhancing generalizability across different tasks [135]. Although only a few studies have attempted to use KGs specifically for HTC, we consider them a promising research direction, given their success in broader TC and NLP applications.

Finally, alternative *tuning* procedures for LMs are being heavily researched in NLP and have showcased promising results in HTC as well. *Prompt-tuning* aims to reduce the difference between the pre-training and fine-tuning phases, wrapping the input in a natural language template and converting the classification task into an MLM task. For instance, an input text $x$ may be wrapped as "$x$ is [MASK]" (hard prompt). Virtual template words may also be used, allowing the LM to both learn to predict the "[MASK]" and tune the virtual template words (soft prompt). A *verbalizer* is then utilized to map the predicted word to the final classification [136], with custom verbalizers for HTC also being developed [137]. On the other hand, *prefix-tuning* only tunes and saves a tiny set of parameters in comparison to a fully fine-tuned method while maintaining comparable performance. To do this, prefix-tuning learns prefix vectors associated with the embedding layer (and, in recent works, each LM layer), while the rest of the LM has its parameters frozen, to fit the target domain better. Specific frameworks for HTC have been developed, training prefix vectors by considering their topological relationship in the hierarchy [138].

### 4.2. Recent Proposals

Tables 1–4 display the results of our search for the latest applications of HTC to the NLP domain. While we do not discuss them, it should be mentioned that other domains of application of HMC have much in common with HTC, and one could also draw inspiration from such works (for example, protein function prediction in functional genomics) [139–142].

**Table 1.** Hierarchical text classification methods. Names for models are reported when authors provide them. ✗ and ✓ stand for code missing and code available, respectively.

| Method | Base | Datasets | Code | Novelty |
|---|---|---|---|---|
| HCCMN [143] | TCN, LSTM, ATT | CRTEXT, Fudan, Sogou, TouTiao | ✗ | Combine TCN with LSTM for extraction of contextual information and temporal features from Chinese text |
| [117] | CNN, LSTM, ATT | RCV1, AAPD, Zhihu-QT | ✗ | Usage of local and global information, new Seq2seq model with CNN encoder and LSTM decoder |
| HLSE [144] | CNN | MeSH | ✗ | Hierarchical label set expansion for label regularization |
| [145] | k-means, SVM | Hamshahri | ✗ | HTC of Persian text with weak supervision using clustering |
| [146] | SVM, MLP | - | ✗ | Mixed deep learning and traditional methods for HTC in the automotive domain |
| WeSHClass [147] | CNN | NYT, AAPD, Yelp | ✓ | Weakly supervised HTC with pseudo-document generation |
| HMC-CAPS [88] | CN | BGC, WOS | ✓ | Compare CapsNets with other NNs |
| HTrans [148] | GRU, ATT | RCV1 | ✗ | Recursive training of LMs on branches of tree-shaped hierarchies of labels |
| NeuralClassifier [149] | - | RCV1, Yelp | ✓ | Toolkit to quickly set up HTC pipelines with multiple algorithms |
| HARNN [76] | RNN, ATT | Patent | ✓ | Hierarchical attention unit (HAM) |
| HiLAP [99] | BASE | RCV1, NYT, Yelp | ✓ | RL framework to learn a label assignment policy |
| HLAN [7] | LSTM, ATT | EURLEX, EURLEX-PT | ✓ | HTC of legal documents, new Portuguese corpus |
| [150] | RF, DT, NB, SVM | HS-Ind | ✓ | Categorization of hate speech expression in Indonesian tweets |
| HKUST [151] | NB, EM | RCV1, 20NG | ✓ | Path cost-sensitive learning algorithm to leverage structural information of hierarchy and unlabeled/weakly-labeled data |
| NETHIC [152,153] | MLP | DANTE | ✓ | DNN for efficient and scalable HTC |
| LSNN [154] | MLP | 20NG, DBpedia, YahooA | ✗ | Unsupervised clustering to exploit label relation, neural architectures to learn inter- and intra-label relations |
| HG-Transformer [111] | TRAN | RCV1, Amazon | ✗ | Transformer-based model, weighted loss using semantic distances |
| H-QBC [155] | QBC (AL) | Enron, Reuters | ✗ | Active learning framework for HTC for classification of datasets with few labeled samples |
| GMLC [118] | LSTM, ATT | - | ✗ | Seq2seq multi-task framework with hierarchical mask matrix to learn cross-task relations |
| 3LBT [156] | SVM, NB, DT, RF | HARD | ✗ | Framework for sentiment classification of Arabic texts using multi-level hierarchical classifiers, using SMOTE technique |
| [157] | | FIN-NEWS-CN | ✗ | Weakly supervised model driven by user-generated keywords, adopting a confidence-enhancing method during training |
| C-HMCNN [158] | MLP | Enron, 19 + | ✓ | Coherent predictions using a hierarchical constraint, usage of hierarchical information |
| HyperIM [110] | GRU | RCV1, Zhihu, WikiLSHTC | ✓ | Embedding of label hierarchy and document in the same hyperbolic space, with explicit modeling of label–word semantics |
| LCN [159] | CNN | 20NG | ✗ | Different approaches with CNNs for HTC |
| [119] | GRU, ATT | WOS, DBpedia | ✗ | Seq2seq model, auxiliary tasks, and beam search |
| HiAGM [97] | LSTM, GCN | RCV1, WOS, NYT | ✓ | Extraction of hierarchical label information |
| ONLSTM [113] | LSTM | WOS, DBpedia | ✓ | Sharing parameters for sequential prediction of next-level label |
| F-HMTC [160] | BERT | - | ✓ | BERT embeddings with hierarchy-based loss |
| [161] | TRAN | B-SHARP | ✗ | Hierarchical Transformer for classification based on ensemble of three models |
| CorNet [162] | BERT, CNN, RNN | EURLEX, Amazon, Wikipedia | ✓ | Model-agnostic architecture for MLTC to enhance label predictions to consider the labels' correlation |
| MAGNET [125] | LSTM, GAT | Reuters, RCV1, AAPD, Slashdot, Toxic | ✓ | Improving GCN with GAT for considering label correlation and co-occurrence |

**Table 2.** Hierarchical text classification methods. Names for models are reported when authors provide them. Base models marked as "n/a" represent works that propose new losses, metrics, or training regimes rather than new methods. ✗ and ✓ stand for code missing and code available, respectively.

| Article | Base | Datasets | Code | Novelty |
|---|---|---|---|---|
| PAC-HCNN [8] | CNN, ATT, GLU | - | ✗ | Stacked hierarchical convolutional layers for HTC of Chinese patent dataset |
| [163] | Co-training | - | ✗ | Semi-supervised approach for HTC of research papers based on co-training algorithm |
| JMAN [164] | GRU | Bibsonomy, Zhihu, CiteULike | ✓ | Attention mechanism to mimic user annotation behavior, loss regularization to capture label co-occurrence relations |
| [120] | LSTM, TRAN | USPTO | ✗ | HTC as sequence generation, teacher forcing with scheduled sampling in training, beam search to choose output labels |
| MATCH [165] | TRAN (CorNet) | MAG-CS, PubMed | ✓ | E2E framework for classification with large hierarchies of labels and documents' metadata |
| [166] | TRAN, GNN | MAG-CS, PubMed | ✗ | Incorporate text metadata and label structure using heterogeneous graphs |
| RLHR [167] | BERT, RL | WOS, Yelp, QCD | ✓ | Label structure modeling using Markov decision process and RL with pre-trained LMs for zero-shot TC |
| HCSM [168] | LSTM, CN | RCV1, EURLEX, WOS | ✗ | Combine global label structure and partial knowledge learning to emulate cognitive process |
| HiMatch [124] | BERT, GCN, GRU, CNN | RCV1, EURLEX, WOS | ✓ | Semantic matching method to learn relationship between labels and text |
| HIDDEN [169] | CNN | RCV1, NYT, Yelp | ✓ | Embed hierarchical label structure in label representation jointly training the classifier |
| HE-AGCRCNN [170] | CNN, ATT, CN, RNN | RCV1, 20NG | ✓ | Architecture, graph-of-words text representation, taxonomy-aware margin loss |
| HVHMC [171] | GCN | WOS, AAPD, Patent | ✗ | Loosely coupled GCN to capture word-to-word, label-to-word, and label-to-label associations, correlation matrix and hybrid algorithm to capture vertical and horizontal dependencies |
| CoPHE [172] | n/a | MIMIC-III | ✓ | New metrics for hierarchical classification for large label spaces |
| CLED [87] | CN, CNN, GRU | DBpedia, WOS | ✓ | Concept-based label embeddings to model sharing in hierarchical classes |
| HMATC [173] | Tree-based (HOMER) | MLAD | ✓ | HTC in multilabel setting for Arabic text, introduction of new Arabic corpus |
| ICD-RE [6] | AE, TRAN | MIMIC | ✓ | Re-ranking method to learn label co-occurrence with unknown label hierarchies |
| SASF [126] | GCN, GRU, CNN | WOS, BGC | ✗ | Hierarchy-aware label encoder and attention mechanism applied to document labels and words |
| L1-L2-3BERT [115] | BERT | - | ✗ | Hierarchical classification using multi-task training with penalty loss |
| HierSeedBTM [174] | HuffPost, 20NG | - | ✗ | Dataless hierarchical classification with iterative propagation mechanism to exploit hierarchical label structure |
| THMM [175] | ATT | Patent | ✗ | Pre-trained LM with multi-task architecture for efficient HTC |
| TaxoClass [176] | RoBERTa | Amazon-531, DBpedia-298 | ✗ | HMTC with class surface name as the only supervision signal |
| HTCInfoMax [177] | HiAGM-LA | RCV1, WOS | ✓ | Text-label mutual information maximization and label prior matching |
| PAAM-HiA-T5 [178] | T5 | RCV1, NYTimes, WOS | ✗ | Hierarchy aware T5, path-adaptive mask mechanism |
| HFT-ONLSTM [113] | LSTM | WOS, DBpedia | ✗ | Hierarchical fine-tuning approach with joint embedding learning based on labels and documents |
| HPT [136] | BERT, GAT | RCV1, WOS, NYT | ✓ | Hierarchy-aware prompt tuning and MLM suitable for handling label imbalance |

**Table 3.** Hierarchical text classification methods. Names for models are reported when authors provide them. Base models marked as "n/a" represent works that propose new losses, metrics, or training regimes rather than new methods. ✗ and ✓ stand for code missing and code available, respectively.

| Article | Base | Datasets | Code | Novelty |
| --- | --- | --- | --- | --- |
| HE-HMTC [179] | BiGRU, SDNE | WOS, Amazon, DBpedia, WebService, BestBuy | ✓ | Level-by-level hybrid embeddings, graph embedding of category structure |
| LA-HCN [94] | ATT | RCV1, Enron, WIPO-alpha, BGC | ✗ | Label-based attention learned at different hierarchical levels, local and global text embedding |
| HGCLR [91] | TRAN | RCV1, WOS, NYT | ✓ | Contrastive learning, graph-based Transformers for sample generation |
| Seq2Tree [93] | T5 | RCV1, WOS, BGC | ✗ | Decoding strategy to ensure consistency in predicted labels across levels, new metrics for HTC |
| HBGL [180] | BERT | RCV1, WOS, NYT | ✓ | Hierarchical label embeddings encoding label structure using both local and global hierarchical data |
| MF-RA [181] | MF, ATT | - | ✗ | Recursive attention mechanism to capture relations between labels at different levels, applied to the MF algorithm |
| CHAMP [107] | n/a | RCV1, NYT | ✗ | Definition of new metrics and modified BCE loss for hierarchical classification |
| OWGC-HMC [114] | MLP | Aizhan | ✗ | Online classifier for genre classification based on entity extraction and embedding with hierarchical regularization term |
| HMC-CLC [182] | BN | RCV1, Enron, imclef07a | ✗ | Exploit label correlations for HTC with greedy label selection method to determine correlated labels dynamically |
| ML-BERT [101] | BERT | Linux Bugs | ✓ | Global approach with multiple BERT models trained on different hierarchy levels, test of document embedding strategies |
| DASHNet [183] | LSTM, GRU | DialogBank, GDC | ✓ | BiTree-LSTM computes prior probabilities of hierarchical label correlation, grammatical, lexical, and contextual encoding |
| P-tuning v2 [138] | BERT | WOS, Amazon | ✓ | Prefix tuning in local and global HTC methods to reduce parameters to fine-tune in pre-trained LMs |
| HR-SciBERT [184] | Bi-LSTM, BERT | WOS, CORA, MAG, SciHTC | ✓ | New dataset, multi-task learning jointly learns topic classification and keyword labeling |
| HLSPC [127] | GCN, Bi-LSTM, ATT | USPTO | ✗ | Leverage label descriptions of patents to find semantic correlation between texts and hierarchical label descriptions with attention |
| TReaderXML [185] | TRAN, ATT | RCV1, Amazon, EURLex | ✗ | Dual cooperative network based on multi-head self-attention, dynamic and fine-grained semantic scope from teacher knowledge to optimize text prior label semantic ranges |
| LD-GGNN [186] | GCN, BiGRU | RCV1, WOS, NYTimes | ✗ | Gated GNN as structure encoder with fine-grained adjacency matrix for better label dependence extraction and reduced over-smoothing, dynamic label dividing to differentiate sibling labels |
| [187] | MLP, BERT | Non-profit Tweets (private) | ✗ | Neighborhood exploration data augmentation, MLP blocks for each layer of the hierarchy with modular adapters |
| [188] | BoW, CNN, TRAN | RCV1, DBpedia, BestBuy | ✓ | Analyze and compare privacy–utility trade-off in DP HTC under a white-box MI adversary |
| LSE-HiAGM [123] | GCN, HiAGM | RCV1, WOS, NYTimes | ✗ | Introduce a common density coefficient of labels to measure their structural similarity, re-balance loss to alleviate label imbalance |
| Seq2Label [121] | BART | RCV1, WOS, BGC | ✗ | HTC as a sequence generation problem, novel Seq2Label framework to learn label hierarchy in a random generative way, hierarchy-aware negative sampling against error accumulation |
| XR-LAT [189] | TRAN | MIMIC-II/III | ✓ | Transformer recursively trained model chain on a predefined hierarchical code tree with dynamic negative label sampling |
| HTC-CLIP [190] | BERT | WOS, NYTimes | ✗ | Combines contrastive learning-guided hierarchy in a text encoder and a path-guided hierarchy |
| GACaps-HTC [191] | GAT, CN | RCV1, WOS | ✓ | A GAT extracts textual representations while integrating label hierarchy information, whereas CN learns latent label relationships to infer classification probabilities |

**Table 4.** Hierarchical text classification methods. Names for models are reported when authors provide them. ✗ and ✓ stand for code missing and code available, respectively.

| Article | Base | Datasets | Code | Novelty |
|---|---|---|---|---|
| HTMC-PGT [192] | XLNet, LSTM | PGT (private) | ✗ | HTC as a parameter-solving problem for multiple multiclass classifiers in a classifier tree |
| [193] | BERT | Private | ✓ | Multi-per-level local classifiers based on BERT for analyses of wildlife exploitation |
| HiDEC [194] | ATT | RCV1, NYTimes, EURLEX | ✓ | HTC as a sub-hierarchy sequence generation, recursive hierarchy encoder–decoder with self-attention to exploit dependency between labels |
| UMP-MG [129] | GCN | E-commerce, WOS | ✗ | HTC as sequence generation through auto-regressive decoder, top-down and down-top unidirectional message-passing |
| Z-STC [195] | BERT | WOS, DBpedia, Amazon | ✓ | Zero-shot prior generation for labels based on semantic alignment of documents and labels, upwards score propagation (USP) to propagate confidence scores upwards in the taxonomy |
| LED [196] | BERT | RCV1, WOS, BGC, AAPD | ✗ | Hierarchy-aware attention module and pairwise matching task to capture hierarchical dependencies of labels |
| HLC-KEPLM [197] | BERT, RoBERTa | Private | ✗ | Knowledge-enhanced pre-trained language model that learns with multiple local classifiers with knowledge fusion |
| [198] | Graphormer, BERT, ATT | RCV1, WOS, NYTimes | ✗ | Generative data augmentation for HTC, leverages both semantic-level and phrase-level hierarchical label information to enhance the label controllability and text diversity |
| [5] | Bi-LSTM | MIMIC-III | ✗ | Two-stage decoding model for hierarchical 2-level ICD codes |
| K-HTC [135] | GCN, BERT | WOS, BGC | ✓ | Knowledge graph integration in text and label representation, contrastive learning with knowledge-aware encoder |
| [128] | BERT, GCN | EDOS SemEval-23 | ✗ | BERT to encode text and labels, hierarchy-relevant structure encoder to model relationships between classes, class-aware attention, self-training against imbalanced classes |
| HypEmo [199] | RoBERTa | GoEmotions, Emp. Dialogues | ✓ | Label embeddings learned in both Euclidean and hyperbolic space to combine LMs with hierarchy-aware representation |
| HiTIN [200] | TextRCNN, BERT | RCV1, WOS, NYTimes | ✓ | Structure encoder optimization for dual-encoder frameworks through the encoding of label hierarchies into unweighted trees |
| LCN [201] | CN | WOS, BGC | ✗ | Hierarchical capsule framework, each capsule classifies one label, same level labels correspond to groups of competing capsules |
| HierVerb [137] | BERT | RCV1, WOS, DBpedia | ✓ | Multi-verbalizer framework, learn vectors as verbalizers constrained by hierarchy and contrastive learning, path-based metric |
| DLAC [202] | Bi-LSTM, CNN | WOS, BGC, AAPD | ✗ | Dual-channel text classification, a global label classification channel, and a deep-level label-assisted classification channel |
| HOOM [9] | SVM | Private | ✗ | IT ticket classification, hybrid model based on SVM-based offline component and Passive Aggressive Classifier online component |
| PeerHTC [130] | GCN | WOS, BGC | ✓ | GCN to reinforce feature sharing among peer (same-level) labels, sample importance learning to alleviate label confusion |
| TML/TARA [203] | Base | WOS, Emotion, GoEmotions | ✓ | Calibration method for prompt-based learning, improve information diffusion and context shift issues in the output embeddings |
| Seq2Gen [122] | mBART | Blog (novel) | ✗ | Unseen label generation by considering label hierarchy as a sequence of labels, semi-supervised learning for better generation |
| MTMD [204] | Bi-LSTM, BERT | Private | ✗ | HTC as sequence generation, multilevel decoupling strategy to guide label generation |
| [205] | GCN | Private | ✗ | GCN-based multilevel classification for users' needs in online medical communities to improve targeted information retrieval |
| HJCL [206] | BERT | RCV1, BGC, NYTimes, AAPD | ✓ | Instance-wise and label-wise contrastive learning techniques on hierarchy |
| [207] | RoBERTa | SemEval 20 T11 | ✓ | Multi-instance multilabel learning for propaganda classification, simultaneously classifies all spans in an article and incorporates the hierarchical label dependencies from the annotation process |

### 4.3. Analyzed Methods

As the number of works is too large to provide an exhaustive description of each of the proposals in Tables 1–4, we provide an analysis of a subset of these based on two

criteria: (*i*) the authors provide a code implementation, and (*ii*) the methods were tested on the two most common public datasets (RCV1 and WOS). These methods are also the ones considered when gathering implementations to test for the experimental part of this survey.

### 4.3.1. HTrans

In their work, Banerjee et al. [148] propose Hierarchical Transfer Learning (HTrans), a framework to improve the performance of a local per-node classification approach. The general intuition is that knowledge may be passed to lower-level classifiers by initializing them with the parameters of their parent classifiers. First, they utilize a bidirectional GRU-based RNN enhanced with an attention mechanism as a text encoder and then use a fully connected network as a decoder to produce the class probability. Word embeddings are initialized with GloVe pre-trained embeddings. One such model is trained for each node in the hierarchy tree with binary output, and child nodes share parameters with the classifiers of ancestor nodes. This can be seen as a "hard" sharing approach, which utilizes fine-tuning to enforce inductive bias from parent to child nodes. The inference is achieved through a standard top-down approach. The authors perform an ablation study by removing parameter sharing and attention and also compare their results with a multilabel model initialized with weights from the binary classifiers, showcasing solid improvements when including their proposed enhancements.

### 4.3.2. HiLAP

Mao et al. [99] tackle the issues that arise from a mismatch between training and inference in local HTC approaches. They propose a reinforcement-learning approach as a solution, modeling the HTC task as a Markov decision process; the task consists in learning a policy that considers where to place an object (which label) as well as when to stop the assignment process (allowing for NMLNP). In other words, such a policy allows the algorithm to "move" between labels or "stop" when necessary. Though theoretically extendable to any neural encoder, the authors use TextCNN [58] with GloVe vectors and BoW features in the document encoder to produce fixed-size embeddings. They also create randomly initialized embeddings for each label, producing an "action embedding" matrix, as each label defines an action for the agent. The encoded document is concatenated with the embedding of the currently assigned label to produce the current state vector. After passing it through a two-layer fully connected network with ReLU activation, the state vector is multiplied by the action embedding matrix to determine the probabilities of all possible actions. In the beginning, each document is placed on the root node, and the assignment stops when the "stop" action is selected as the most probable. The loss function is defined in terms of the overall $F_1$ score between the previous and the current time step. The proposed Hierarchical Label Assignment Policy (HiLAP) yields excellent results, in particular in terms of consistency of parent-child assignments concerning the hierarchy of classes.

### 4.3.3. MATCH

The authors of MATCH [165] propose to boost the multilabel classification performance by learning a text representation enriched with document metadata and further adding a regularization objective to exploit the label hierarchy. The first component of their architecture is a metadata-aware pre-training scheme that jointly learns words and metadata embeddings considering the vicinity between documents and related metadata and labels. A modified Transformer encoder is then used to compute document representations: to cope with large label spaces, several special tokens (`[CLS]`) are pre-pended to input sequences, followed by the metadata tokens and the document's words. The representation of all `[CLS]` tokens is then concatenated and passed into a fully connected layer with a sigmoid activation function. An L2 regularization (hypernymy regularization) is applied to the classification layer (i.e., to the weight matrix), forcing the parameters of each label to be

similar to those of its parent. For all pairs of parent ($l$) and child labels ($l'$) the regularization term is expressed as:

$$R_{\text{parameter}} = \sum_{l \in \mathcal{L}} \sum_{l' \in \sigma(l)} \frac{1}{2} \|\mathbf{w}_l - \mathbf{w}_{l'}\|^2 \tag{20}$$

where $\mathcal{L}$ is the set of labels, $\sigma(l)$ is the set of parent labels of $l$, and $\mathbf{w}_l$ denotes the parameters of a label $l$. The final predictions are also regularized to penalize the model when the probability of a parent label is smaller than the one for the child label. The value is summed over all documents $d \in \mathcal{D}$, as in the equation below:

$$R_{\text{output}} = \sum_{d \in \mathcal{D}} \sum_{l \in \mathcal{L}} \sum_{l' \in \sigma(l)} \max(0, \pi_{dl} - \pi_{dl'}) \tag{21}$$

where $\pi_{dl}$ represents the probability that document $d$ belongs to child class $l$. The authors report the results of an ablation study that confirms the metadata embedding strategy and hierarchy-aware regularization are both beneficial to the classification task.

### 4.3.4. HiAGM

Zhou et al. [97] propose an end-to-end hierarchy-aware global model (HiAGM) that leverages fine-grained hierarchy information and aggregates label-wise text features. Intuitively, they aim to add information to traditional text encoders by introducing a hierarchy-aware structure encoder (the structure being the hierarchy). As structure encoders, the authors test a TreeLSTM and a GCN adapted to hierarchical structures. Moreover, they propose two different frameworks: one based on multilabel attention (HiAGM-LA), and one on text feature propagation (HiAGM-TP). HiAGM-LA utilizes the attention mechanism to enhance label representations in a bidirectional, hierarchical fashion, utilizing node outputs as the hierarchy-aware label representation. HiAGM-TP, on the other hand, is based on text feature propagation in a serial dataflow; text features are used as direct inputs to the structure encoder, propagating information throughout the hierarchy. For multilabel classification, the binary cross entropy (BCE) loss is used, as well as the regularization term $R_{\text{parameter}}$ that was described for MATCH.

### 4.3.5. RLHR

The approach used by Liu et al. [167], which tackles zero-shot HTC, includes a reinforcement learning (RL) agent that is trained to generate deduction paths, i.e., the possible paths from the root label to a child label, to introduce hierarchy reasoning. The reward is assigned depending on the correctness of the predicted paths, which should be sub-paths of the ground truth set of paths to positively reward the agent. Moreover, the authors design a rollback algorithm that overcomes the inefficiencies of previous solutions and allows the model to correct inconsistencies in the predicted set of labels at inference time. The zero-shot task is formulated as a deterministic Markov decision process over label hierarchies. BERT and DistilBERT are used as base models, which are pre-trained on a binary classification task with a negative sampling strategy. Several training examples are created by pairing each document with one of its labels as well as some irrelevant labels to provide positive and negative examples. Using the pre-trained model, a policy is learned to further tune the model on the binary classification task.

### 4.3.6. HiMatch

In HiMatch [124], the HTC task is framed as a semantic matching problem, and the method is used to jointly learn embeddings for documents and labels as well as to learn a metric for matching them. The proposed architecture first utilizes a text encoder akin to the one used by HiAGM. Then, a *label encoder* is used to produce label embeddings enriched with dependencies among labels. It uses the same GCN architecture as the text encoder; label vectors are initialized with BERT's pre-trained embeddings. The document and label representations are used in the label semantic matching component, which projects text

and labels into a common feature space using two independent two-layer feed-forward networks. A cross-entropy objective is used for training with two regularization terms: one to force documents and respective labels to share a similar representation, measured in terms of mean square error, and a second to penalize close semantic distance between a document and incorrect labels. The latter constraint uses a triplet margin loss objective with Euclidean distance. All components are trained jointly, and the authors report improved performance over a BERT-based model fine-tuned on multilabel classification.

### 4.3.7. HE-AGCRCNN

The usage of CNs has also been proposed for HTC tasks. Aly et al. [88], for instance, adapt a CN to exploit the labels' co-occurrence, correcting the final predictions to include all ancestors of a predicted label to ensure consistent predictions. More recently, Peng et al. [170] discuss the drawbacks and strengths of several popular neural networks used for text processing, including CNNs, RNNs, GCNs, and CNs. They propose to combine them in a single architecture (AGCRCNN) so that they can better capture long- and short-range dependencies and both sequential and non-consecutive semantics. In their work, documents are modeled using a graph of words that retains word ordering information: after a lemmatization and stop-word removal step, each word is represented as a node with its position in the document set as an attribute. A sliding window is passed over the document and edges between nodes are created, weighted on the number of times a word co-occurred in all sliding windows. For each document, they extract a sub-graph whose nodes are the document's words with the highest *closeness centrality*—a measure of the importance of words in a document—and their neighbor nodes, up to a fixed number. The nodes are then mapped to Word2Vec embeddings, obtaining a 3D representation that is fed to the attentional capsule recurrent CNN module. This is composed of two blocks, each one containing a convolutional layer to learn higher-level semantic features, and an LSTM layer with an attention mechanism to learn local sequentiality features specific to each sub-graph. A CN with a dynamic routing mechanism is finally used for multilabel classification. They perform an ablation study comparing several variations of their architecture, as well as previously proposed deep learning classifiers, showcasing better results for their proposed model on two datasets.

### 4.3.8. CLED

As another example of the application of CNs to HTC, Wang et al. [87] propose the Concept-based Label Embedding via Dynamic routing (CLED), in which a CN is used to extract concepts from text documents. Concepts can be shared between parent and child classes and can thus be used to support classification based on hierarchical relations. The top-$n$ keywords from each document are used as concepts and encoded with GloVe word embeddings. A clustering procedure is utilized to initialize concepts' embeddings with the clusters' center. Dynamic routing is then used to learn concepts' embeddings; agreement is only measured between capsules representing parent and child classes.

### 4.3.9. ICD-Reranking

Tsai et al. [6] tackle the task of automatic ICD coding in medical settings, which requires multilabel classification of clinical nodes in hierarchically dependent diagnostic codes. The authors propose to pair a base predictor (responsible for the generation of top-$k$ most probable label sets) with a re-ranking step. In particular, it is the re-ranker that is designed to capture correlation and co-occurrence between labels. They propose two agnostic re-ranking methods, which they validate across different base predictors to prove the generalizability of their proposed re-rankers. Both re-ranking methods are trained on the same training data as the candidate generator. The first approach, MADE, uses joint probability as a way to score label sets, estimating it by decomposing it in an autoregressive fashion with random ordering. As this approach may fail to capture non-sequential dependencies, they also propose Mask-SA, a self-attentive approach inspired

by NLP's masked language models (MLM) that estimates a distribution over the label vocabulary for the masked input given all other elements in the set. The authors showcase a consistent improvement across three different predictors on two datasets.

### 4.3.10. HGCLR

Wang et al. [91] argue against the encoding of text and label hierarchy separately, and instead propose to aggregate the two representations. They emphasize the fact that the label hierarchy is static, thus translating to an individual representation by the graph encoder. As the interaction becomes redundant, they argue for the direct injection of this representation into the text encoder. The hierarchy-aware representation is achieved through a contrastive learning approach focused on generating positive examples that are both label-guided and hierarchy-involved. The construction of such examples is driven by the observation that a select number of keywords is sufficient to attach a label; similar to an adversarial attack, then, new examples might be created by modifying tokens within the text. However, the aim is not to disrupt the example like in an adversarial attack but rather to modify unimportant tokens to keep the classification results unchanged. From a technical perspective, important keywords are defined by evaluating the attention weight of token embeddings, while graph embeddings are obtained through a modified Graphormer [208].

### 4.3.11. HIDDEN

Chatterjee et al. [169] devise a way to create label embeddings by leveraging the properties of hyperbolic geometry, which is helpful in the representation of organized structures (like hierarchies). They adopt a specific hyperbolic model, the Poincaré ball model. Briefly, in this model, the distance between pairs of points falls exponentially as one moves from the origin toward the surface of the ball. The authors claim that this property can be used to represent arbitrarily large hierarchies, where the root of the hierarchy can be thought of as being close to the origin and the leaves lie on the surface of the ball. A TextCNN [58] is used to learn document embeddings, while the hyperbolic model is used to learn label embeddings through a document–label alignment criterion based on the dot product between embeddings. Additionally, a second loss term is used to push labels closer in the hyperbolic embedding space based on their co-occurrence. Their results suggest that the joint learning objective increases performance metrics compared to a model sequentially optimized on the two objectives.

### 4.3.12. CHAMP

Vaswani et al. [107] propose a modified BCE loss definition that accounts for the severity of mispredictions, which they name Comprehensive Hierarchy Aware Multilabel Predictions (CHAMP). More specifically, the CHAMP loss function penalizes false positives depending on the distance between the incorrectly predicted label and the true labels. On the other hand, false negatives are always considered equally severe. For a set of labels $L$, a ground truth vector $y \in \{0,1\}^L$, and a prediction $\hat{y} \in \{0,1\}^L$, the loss function is defined as:

$$\text{CHAMP}(y, \hat{y}) = -\sum_{j=1}^{|L|} y_j \log \hat{y}_j + (1 + s_S(j))(1 - y_j) \log(1 - \hat{y}_j) \quad (22)$$

where $s_S(j)$ is the measure of severity of false positive label $j$, which depends on the distance from the ground truth label set $S$. Assuming $\text{dist}(j, S)$ is the minimum distance between $j$ and any label in $S$, the severity can be defined as:

$$s_S(j) = \beta \cdot \frac{\text{dist}(j, S)}{\text{dist}_{max}} \quad (23)$$

Being an alternative loss function, this approach is natively model-agnostic. The authors found substantial improvements in terms of the area under the precision–recall

curve (AUPRC) and retrieval metrics over their baselines trained using the BCE loss on several datasets.

### 4.3.13. HE-HMTC

Ma et al. [179] propose HE-HMTC, a local per-level approach where a different model is used at each level of the hierarchy of labels. First, a bidirectional GRU encoder is used for text representation. Forward and backward hidden states are concatenated to obtain an encoded document. A Structural Deep Network Embedding (SDNE) [209] graph embedding method is used to obtain compressed representations for each label in the hierarchy. Specifically, SDNE is an auto-encoder trained separately to reconstruct the adjacency matrix of the DAG that is used to represent the hierarchy and allows the capture of both structural and semantic features of the label set. At each level, the classifier receives the text representation from the encoder at the previous level, as well as the previous label embedding. This vector is finally passed through a fully connected layer and a softmax activation, generating the final probabilities for a level of classes. The process stops when the last level of labels is reached. They validate the model on several datasets and perform an ablation study to confirm the impact of the graph encoder and different text encoding schemes.

### 4.3.14. HTCInfoMax

Deng et al. [177] build upon the work carried out by Zhou et al. [97], adding two modules on top of the HiAGM model. They use the probability distribution produced by the encoder and a discriminator to estimate the mutual information between a document and its labels. This module is used to clean documents of irrelevant label information and to embed documents with corresponding labels' information. The second module is used to constrain the label encoder to learn better label representations, especially for low-frequency labels. They use a loss function to maximize the mutual information between text and labels, as well as a regularization term that pushes the learned distribution of labels close to its true distribution. The authors showcase superior performance to HiAGM on two datasets, as well as demonstrating the effectiveness of each module through an ablation study.

### 4.3.15. GACaps-HTC

Bang et al. [191] propose a global approach combining GNNs to extract and encode label-hierarchy information and CNs to learn implicit relationships between labels. The GACaps-HTC first encodes documents using pre-trained BERT or SciBERT models, followed by a convolutional layer to generate label-specific embeddings of the input text for each label. These representations are then used in a GAT to embed hierarchy information through the propagation of label-specific information to neighbor labels. The label-specific and hierarchy-aware representations are generated based on the attention weights learned by the GAT and are finally passed to a two-layer CN. The activation vectors from primary capsules determine the probability of labels for each specific document and the dynamic routing algorithm learns how to distribute this information to the digit capsules that perform the final classification. The model is optimized end-to-end using focal loss and a contradiction term is added to encourage the model to label with fine-grained labels only when the parent label is also assigned. The results are reported to be superior to HGCLR (the strongest baseline) [91] on both the WOS and RCV1 datasets.

### 4.3.16. HiDEC

Im et al. [194] propose an approach to reduce the complexities of hierarchy-aware models, pairing a hierarchy encoding mechanism with a text-hierarchy attention mechanism. First, a minimal sub-hierarchy is extracted for each document containing its target labels and encoded as a sequence with positional encoding. The authors assume that it is not needed to consider the whole hierarchy since most of the labels are irrelevant to a

document. Subsequently, hierarchy embeddings are processed using multi-head masked self-attention to learn an encoding for the sub-hierarchy sequences. Input documents are embedded using a text encoder. A multi-head attention mechanism is used to match hierarchy information to encoded documents, producing a weighted representation of tokens to the document labels. Finally, the recursive decoder uses this representation to decode the hierarchy of labels of a document. The model is trained end-to-end using a modified BCE loss, and strong results are reported on the RCV1 dataset, beating baselines like HGCLR and HiMATCH.

### 4.3.17. K-HTC

Liu et al. [135] leverage external knowledge from ConceptNet (a knowledge graph) to improve HTC. First, input documents are processed to recognize and map concepts to ConceptNet entities. These entities and their relations are extracted to compose a pruned KG, which is used to pre-train concept embeddings utilizing the TransE model [133]. The pre-trained concept embeddings are then passed to a knowledge-aware text encoder based on GraphSAGE [210] to obtain expressive node representations for each concept that also consider neighboring concepts. Raw word embeddings are obtained with BERT and then fused with the representation from aligned concepts. The same encoding strategy is also used to obtain knowledge-aware embeddings of target labels, which can be seen as an acyclic graph. A GCN is then used to propagate the representation of labels on the label hierarchy graph. Finally, label attention is utilized to obtain the class-enhanced document representation. BCE loss is used in the optimization, along with two contrastive loss terms. The first ensures that document representations are closer when the number of concepts shared by the document is high. Similarly, the second moves documents with related labels (i.e., with shared ancestors in the hierarchy tree) closer. Experiments show strong performance, with results superior to HGCLR and HPT.

### 4.3.18. HiTin

Zhu et al. [200] propose to convert the label hierarchy into a simplified tree structure to remove noisy (or less discriminative) information and then encode this knowledge in the text representations. First, text is encoded using BERT (though any text encoder would work). Then, a structure encoder applies the complexity reduction to the hierarchy of labels through an entropy-minimization scheme, which produces the simplified hierarchy tree. A message-passing mechanism is then adopted to learn document representation based on the latter. A BCE loss is used, with a regularization term to encourage related labels to share model parameters (soft parameter sharing). Experiments show strong performance, with results superior to HGCLR, HiMATCH, and other BERT-based models.

### 4.3.19. PeerHTC

Song et al. [130] argue that labels in different branches of the hierarchy may still carry mutual relevancy information and thus develop an approach that aims to exploit such latent relationships. To encode more information in the label embeddings, two strategies are used. The first strategy is similar to HiAGM and is used to embed hierarchy information (depth-wise), combining top-down and bottom-up embeddings obtained by an LSTM encoder. Breadth-wise relationships (encoded in peer-wise embeddings) are also learned using two GCNs, one considering only labels at the same depth and one considering the complete hierarchy. Secondly, the authors propose a strategy to learn the adjacency matrix used in the GCN. To do this, a warm-up training phase is utilized to initialize the adjacency matrix used in the GCN for the training phase. Two approaches are tested, using Spearman's correlation and cosine similarity between predicted probabilities and embeddings, respectively. The final hierarchy- and peer-aware embeddings are then fused using a non-linear projection to form the final label embeddings. The attention mechanism is used to align this label information with the encoded documents and is used to produce the final weighted representation for a document that is used for classification.

The BCE loss is used with a weighting strategy to penalize confusion between labels that frequently appear together in the dataset. The reported results surpass the performance of HTCInfoMax and HiAGM baselines.

### 4.3.20. HJCL

Yu et al. [206] propose a contrastive learning approach to overcome the challenges of previous work in determining contrastive pairs of samples. Both labels and documents are embedded using BERT, and a GAT is used to refine the label embeddings to reflect hierarchical information. Transformer's MHA is then used to obtain label-aware embeddings weighted by their relatedness to each label. Contrastive learning is then used to narrow the distance between representations of anchors with positive labels, with the added constraint that lower-level labels should be closer than higher-level ones. However, this approach reportedly does not perform well in a multilabel setting, as labels that are close in the hierarchy could be moved far apart. Hence, a contrastive learning step between labels is also added to ensure that label embeddings with the same target labels are kept close in the latent space. The zero-bounded log-sum-exp and pairwise rank-based loss (ZLPR) [211] is used for optimization, along with the two contrastive learning terms. The results are compared with several baselines, including HGCLR, Seq2Tree, and HTCInfoMax, reporting 2–5% improvements in F1 metrics.

### 4.3.21. HBGL

Jiang et al. [180] devise a method to make use of the global hierarchy and also exploit the local information that is specific to the label hierarchy of each document. The authors use BERT to initialize label representation of the hierarchy graph; attention masks are used to ensure the model attends only to parent and child labels and an MLM objective is used to embed hierarchy information fine-tuning label representations by learning to fill masked leaf labels. To avoid overfitting, the parameters of the BERT encoder are frozen, and only the last layer producing the global hierarchy-aware embeddings is learned. These are used to compute embeddings for local hierarchies: all paths to leaf labels for each example are encoded into a sequence of embeddings and are considered local hierarchies. BERT is trained to predict the next label in the local sequence given the previous labels and the input text. Masking is used to ensure there is no leakage of "target" information, as in the original BERT. At inference time, the model is used in an autoregressive fashion, and the probabilities of each label are computed with a sigmoid activation. Experiments are carried out on three datasets, showcasing improved performance over HGCLR, HTCInfoMax, and HiAGM.

### 4.3.22. HPT

Prompt-tuning approaches aim to close the gap between the pre-training and fine-tuning phases, with many authors claiming that traditional fine-tuning may restrict LMs' capabilities. Wang et al. [136] propose a prompt-tuning approach using an MLM objective for HTC. They experiment with soft and hard prompt strategies, with the former adding virtual tokens to the input sentence for automatic prompt learning. To incorporate hierarchy, the prompt introduces a virtual token for each hierarchy level. These tokens are initialized using BERT word embeddings of the label names. A GAT is used to learn embeddings reflecting the relation between labels, and these are used to initialize the virtual token embeddings. The cross-entropy loss originally used in MLM is not suitable here, since multiple labels must be predicted. Following previous work, the authors use the zero-bounded multilabel cross-entropy loss, which effectively adapts cross-entropy to a multilabel setting, encouraging target labels to score higher than the other labels. Additionally, 15% of tokens from the input text are masked and the standard MLM loss used in BERT is also adopted. The authors report improved results over several baselines, including HGCLR, HiMATCH, and hard and soft tuning using BERT without their hierarchy injection strategy.

### 4.3.23. HierVerb

Ji et al. [137] describe a prompt-tuning approach for few-shot HTC. Similarly to HPT [136], they exploit prompt-based learning with language models and integrate the hierarchical information into the prompt verbalizer using a contrastive learning loss. The prompt with the masked label information and the original text is encoded using BERT, and then a multi-level verbalizer is used, where each level is responsible for learning a representation for masked labels at different depth levels. Two loss terms are added, one to better adapt the pre-trained LM to the hierarchical objective, and a contrastive one to increase the similarity of the learned representation for intra-pair samples. The results are compared with state-of-the-art models, like HGCLR, HiMatch, and HPT, showing competitive performance in both few-shot and full-shot settings.

### 4.3.24. P-Tuning-v2

Chen et al. [138] investigate the usage of prefix tuning for HTC. As mentioned, this technique only learns short vectors (soft prefix prompts, SPP) instead of fine-tuning the entire LM, which has its weights frozen instead. In the paper, SPP vectors are appended to the input representation of each layer, and several tuning strategies are employed. These include a local tuning strategy where a separate model is trained for each level of the hierarchy, a global approach, and a second global approach enhanced with contrastive learning to improve the learned representations. In the local approach, the authors also test different strategies to condition SPP vectors in adjacent hierarchy layers to reflect the hierarchical nature of labels. They use BERT as the pre-trained language model and experiment in multilabel settings with BCE loss. The results are compared among the tested variants on three datasets.

### *4.4. Datasets Used in the HTC Literature*

We conclude this section on the HTC literature by providing an overview of the most commonly utilized datasets in the context of HTC, as inferred by analyzing recent methods in the previous sections. Table 5 summarizes the datasets that are well-defined for HTC tasks and are often encountered in the literature. Table 6 lists large collections of documents from which HTC datasets are often derived, though often inconsistently across different works.

Indeed, much of the HTC literature is spread across a wide variety of datasets, which often collect data from the same source but utilize it in different ways. While what we report in Table 6 are among the most prominent sources in terms of raw data, there are others, such as data derived from DBpedia [212] and Wikipedia [213] dumps. Overall, methods listed in Tables 1–4 span across more than 44 different datasets, many of which are only tested on the specific method being proposed, at least recently. Therefore, HTC suffers from a lack of established benchmarks, resulting in this scattering of methods over a wide range of incomparable datasets.

There is also another source of inconsistency across results in the literature that should be mentioned. Some of these datasets provide pre-defined splits, such as the very popular RCV1 dataset (which is by far the most utilized dataset, followed by the WOS dataset). However, when comparing results with methods, one should be mindful to check whether the authors have made use of such splits; indeed, the most common split of the RCV1 dataset utilizes a very small portion of the overall data for training (around 3%). With this in mind, methods that adopt larger training splits (which is indeed the case in some works) should be compared to others with due precaution.

**Table 5.** Commonly utilized datasets in the HTC literature.

| Name | Size | Depth | Labels (Overall) | Labels per Level |
|---|---|---|---|---|
| RCV1-v2 [214] | 804,414 | 4 | 103 | 4–55–43–1 |
| Web of Science (WOS-46,985) [215] | 46,985 | 2 | 145 | 7–138 |
| Blurb Genre Collection (BGC) [88] | 91,894 | 4 | 146 | 7–46–77–16 |
| 20Newsgroup (20NG) [216] | 18,846 | 2 | 20 | 6–20 |
| Arxiv Academic Paper (AAPD) [171,217] | 55,840 | 2 | 61 | 9–52 |
| Enron [94,218] | 1648 | 3 | 56 | 3–40–13 |
| Patent/USPTO [76] | 100,000 | 4 | 9162 | 9–128–661–8364 |

**Table 6.** Large collections from which HTC datasets are often derived.

| Name | Size | Depth (Overall) | Labels (Overall) |
|---|---|---|---|
| New York Times (NYT) [219] | $\sim$ 1.8 M | 10 | 2318 |
| YELP [99,169] | $\sim$ 7 M | 2 | 539 |
| Amazon [220] | $\sim$ 35 M | 3 | 531 |

## 5. Experiments and Analysis

This section presents the experimental part of this work: first, we describe our selection of benchmark datasets, then we proceed to introduce the baseline methods we tested, and finally, we discuss and compare their performance.

### 5.1. Datasets Used

In our experiments, we select three of the most popular datasets for HTC, namely, the Web of Science [215], Blurb Genre Collection [88], and Reuters Corpus-V1 [214]. The latter two are distributed with pre-defined training and test splits, allowing us to directly compare our results with those of other works. To diversify the domains being tested, we additionally test methods on two more datasets, the first being a collection of bug reports crawled online, and the second a corpus of user reviews that we newly derive from the Amazon corpus. Overall, these datasets are representative of five different domains of applications of HTC methods (i.e., books, the scientific literature, news, IT tickets, and reviews), hence providing us with results across a wide spectrum of diverse data. Statistics for the five datasets are displayed in Table 7, along with an indication of whether they respect the MLNP assumption. The specific preprocessing procedure for each method we tested is discussed in the following sections, along with the experimental details.

**Table 7.** Statistics for datasets used in this work. ✓ and ✗ in the MNLP row stand for mandatory and non-mandatory leaf prediction, respectively. Values for splits are indicated as "n/a" if no standard split exists in the literature.

| | Bugs | RCV1-v2 | WOS | BGC | Amazon |
|---|---|---|---|---|---|
| Size | 35,050 | 804,414 | 46,960 | 91,894 | 500,000 |
| Depth | 2 | 4 ** | 2 | 4 | 2 |
| Labels overall | 102 | 103 * | 145 | 146 | 30 |
| Labels per level | 17–85 | 4–55–43–1 | 7–138 | 7–46–77–16 | 5–25 |
| Average # characters | 2026 | 1378 | 1376 | 996 | 2194 |
| Train | 18,692 | 23,149 | 31,306 | 58,715 | 266,666 |
| Validation | 4674 | n/a | 6262 | 14,785 | 66,667 |
| Test | 11,684 | 781,265 | 15,654 | 18,394 | 166,667 |
| MLNP | ✓ | ✗ | ✓ | ✗ | ✓ |

* Overall, only 101 are available in the training split. ** Removing the unassigned categories.

### 5.1.1. Linux Bugs

The Linux Bugs dataset (which we will refer to simply as "Bugs") was introduced by Lyubinets et al. [221] and comprises bugs scraped from the Linux kernel bugtracker

([https://bugzilla.kernel.org](https://bugzilla.kernel.org), accessed on 17 March 2024). We utilize the script provided by the original authors to acquire a larger set of data. The documents are essentially support tickets classified in terms of importance, related product, and specific components. The "product" field acts as a parent label to the "component" sub-labels, from which we can derive a hierarchy of labels. As the depth of the hierarchy is only two, it is a rather shallow and wide structure. Furthermore, the dataset itself is strongly unbalanced; therefore, we discard bug reports tagged with labels or sub-labels appearing less than 100 times. The resulting dataset is still unbalanced, but this process helps to filter out lesser categories that are barely represented.

In terms of textual content, this dataset is very noisy. Entries often contain grammatical inconsistencies and technical jargon, as well as technical readings such as stack traces or memory addresses. An example of a bug report from this dataset is given in Listing 1.

**Listing 1.** Linux bugs extracted from the Bugs dataset.

```
Description: ''exact kernel version:linux-2.5.51 distribution:redhat 8.0 + linux2.5.51
    hardware environment:intel stl2 mother boar d problem description: compile e100 as
    kernel module, insmod e100 and start the network. then stop network and remove e100
    together. then kernel crashes in random places. for example: use command : insmod e100
    /etc/init.d/network start /etc/init.d/network stop; rmmod e100 then the kernel crashes.
    eflags: 00010887 eip is at cascade + 0x25/0x60 eax: defd02b8 ebx: 00000001 ecx:
    00000000 edx: c150a4c0 esi: c150acd4 edi: c150acd4 ebp: c150acd4 esp: c0559f1c ds: 0068
    es: 0068 ss: 0068 process swapper (pid: 0, threadinfo=c0558000 task=c0497f60) [...]''
Categories: ''Drivers'', ''Networks''
```

5.1.2. RCV1-v2

The Reuters Corpus Volume I (RCV1) dataset [214] is a human-labeled newswire collection of Reuters News collected between 20 August 1996 and 19 August 1997. It contains over 800,000 manually categorized international newswire stories in English. In particular, we adopt the widely utilized corrected version, referred to as RCV1-v2. This version contains several fixes for the categories assigned to each document, while 13 topic codes are removed entirely. We follow the instructions from Lewis et al. [214] to generate the labeled dataset: all articles published from 20 August 1996 to 31 August 1996 are used as the training split, while all the remaining articles (up to 19 August 1997) are used as the test set. As a result of this chronological split, we noticed that two of the topic codes are only present in the testing set. We remove these codes, which leaves us with 101 topics in both training and testing. In our dataset, the headline and article fields of each news article are concatenated to generate the final document. The hierarchy among topic codes is specified in the file `rcv1.topics.hier.expanded`, which can be downloaded from the official RCV1-v2 repository. An example of an article in XML extracted from RCV1-v2 is shown in Listing 2.

**Listing 2.** A news article extracted from the RCV1 dataset.

```
<newsitem itemid=''2307'' id=''root'' date=''1996-08-20'' xml:lang=''en''>
    <title>UK: Oil prices slip as refiners shop for bargains.</title>
    <headline>Oil prices slip as refiners shop for bargains.</headline>
    <dateline>LONDON 1996-08-20</dateline>
    <text>
        <p>World oil prices slipped on Tuesday in a market where refiners stung by high
            crude premiums and poor margins began to bargain for a cheaper barrel.</p>
        <p>October futures for world benchmark Brent Blend crude from the North Sea closed
            down 38 cents at $20.43 a barrel after failing to break through the day's high
            of $20.80.</p>
        <p>&quot;There was a broad feeling in the market that Brent was overheated and had
            to come down,&quot; a trader said.</p>
        <p>On the unofficial Brent forward market, prompt differentials for Dated or
            physical Brent shrank, suggesting cargoes would fetch lower premiums in the
            weeks ahead. This could avert the risk of refineries running less crude through
            their systems to pump up the price of products.</p>
        [...]
```

```
    </text>
    <copyright>(c) Reuters Limited 1996</copyright>
    <metadata>
        <codes class=''bip:countries:1.0''> [...] </codes>
        <codes class=''bip:topics:1.0''>
          <code code=''M14''> [...] </code>
          <code code=''M143''> [...] </code>
          <code code=''MCAT''> [...] </code>
        </codes>
    </metadata>
</newsitem>
```

### 5.1.3. Web of Science

The Web Of Science (WOS) dataset, first introduced by Kowsari et al. [61], contains abstracts from papers published on the Web of Science (https://www.webofscience.com, accessed on 17 March 2024) platform. We utilize the dataset in its complete version (sometimes referred to as WOS-46985 (https://data.mendeley.com/datasets/9rw3vkcfy4 /6, accessed on 17 March 2024), which comprises 46,985 abstracts from published papers in seven major scientific domains. The categories are further subdivided into 134 sub-domains. Much like the Linux Bugs dataset, it is characterized by a shallow hierarchy and an unnatural balancing—given by the fact that each example has exactly two labels [88]. An example of an abstract from the WOS dataset is shown in Listing 3.

**Listing 3.** Abstract extracted from the WOS dataset.

```
Abstract: ''(T)his paper presents the concept of a software-defined radio with a flexible
    RF front end. The design and architecture of this system, as well as possible
    application examples will be explained. One specific scenario is the operation in
    maritime frequency bands. A well-known service is the Automatic Identification System
    (AIS), which has been captured by the DLR mission AISat, and will be chosen as a
    maritime application example. The results of an embedded solution for AIS on the SDR
    platform are presented in this paper. Since there is an increasing request for more
    performance on maritime radio bands, services like AIS will be enhanced by the
    International Association of Marine Aids to Navigation and Lighthouse Authorities
    (IALA). The new VHF Data Exchange Service (VDES) shall implement a dedicated satellite
    link. This paper describes that the SDR with a flexible RF front end can be used as a
    technology demonstration platform for this upcoming data exchange service.''
Categories: ''ECE'', ''Distributed-computing''
```

### 5.1.4. Blurb Genre Collection

The Blurb Genre Collection (BGC) (https://www.inf.uni-hamburg.de/en/inst/ab/ lt/resources/data/blurb-genre-collection.html, accessed on 17 March 2024) was first introduced by Aly et al. [88] and is primarily comprised of so-called "blurbs" (i.e., short advertising texts for books) in English, as well as other metadata such as author name, publication date, and so on. The data are crawled from the Penguin Random House website and preprocessed to remove uninformative categories and category combinations that appear less than five times. As described by the authors, the procedure followed aims at mimicking the properties of the RCV1 dataset, ultimately generating a forest-like hierarchy structure. The label distribution is nonetheless unbalanced, with 146 overall labels and a hierarchy of depth 4. An example of a book summary extracted from this dataset is shown in Listing 4.

**Listing 4.** Book sample extracted from the BGC dataset.

```
<book date=''2018-08-18'' xml:lang=''en''>
    <title>Creatures of the Night (Second Edition)</title>
    <body>Two of literary comics modern masters present a pair of magical and disturbing
        stories of strange creatures who are not quite what they seem! In The Price, a
        mysterious feline engages in a nightly conflict with an unseen, vicious foe. The
        Daughter of Owls recounts an eerie tale of a beautiful orphan girl who was found
        clutching an owl pellet-and how those who would do her wrong would face bizarre,
```

```
         unforeseen consequences. Neil Gaiman (The Sandman, American Gods) delivers his
         award-winning magic and mystery, realized in Michael Zulli's lavish paintings,
         newly re-designed in a beautiful new edition!</body>
     <copyright>(c) Penguin Random House</copyright>
     <metadata>
         <topics>
             <d0>Fiction</d0>
             <d1>Graphic Novels & Manga</d1>
         </topics>
         <author>Neil Gaiman</author>
         <published>Nov 29, 2016 </published>
         <page_num> 48 Pages</page_num>
         <isbn>9781506700250</isbn>
         <url> [...] </url>
     </metadata>
</book>
```

### 5.1.5. Amazon 5 × 5

We synthesize a new dataset composed of user reviews labeled with the two-level category of the reviewed product. The dataset is obtained from the 2018 Amazon dump proposed in Ni et al. [222]. We extract product reviews for the following five categories: "Arts, Crafts and Sewing", "Electronics", "Grocery and Gourmet Food", "Musical Instruments", and "Video Games". Then, we manually exactly five sub-categories for each macro-category. In some cases, we map several sub-categories to the same macro-category (e.g., "Needlework" and "Sewing" are related; hence, grouped in "Sewing") and fix inconsistencies as best we can (e.g., we list "Marshmallows" under the "Candy" category, instead of under the "Cooking & Baking" one). Reviews are sampled in such a way that exactly 100,000 reviews are extracted for each domain. The dataset is therefore balanced concerning the five macro-categories. The second-level labels we obtain are also fairly balanced, though an equal split for each one is not ensured. Labels and sub-labels form a tree hierarchy, and there are exactly two labels for each sample. Reviews are the concatenation of a summary and a longer description, although sometimes only one of these is available. Before extraction, reviews are sorted by length to ensure the longer ones are included in our dataset. A sample review extracted from this dataset is shown in Listing 5.

**Listing 5.** An example of user review in the Amazon dataset.

```
{
   'summary': ''Impressed for $10'',
   'reviewText': ''I picked up an old set of VW recaro bucket seats (the old gray ones).
      One of the cussions had a few holes in them (cigarette burn) so I decided I'd see
      what I could do. I worked on the seat in pieces so I did it inside. It could be
      done just as easily installed in the car. Simply use the color chart to mix a close
      match to the interior. Clean the area you will be working with. If necessary
      provide some backing such as a cotton ball stuffed down inside. [...] I applied the
      fix to several holes about 5 h ago to writing this review and its already setting
      up nicely. I'm going to clear the excess coloring tomorrow and reassemble the seat
      and install them back in my car. Bottom line $10 and i could patch another 10 holes
      with moderately professional results. Can't beat it.'',
   'category': [''Arts Crafts and Sewing'', ''Crafting'']
}
```

### 5.2. Models Implemented

In this section, we provide some technical details on the methods we tested firsthand. We select as candidates the methods described in Section 4.3, i.e., those providing results on RCV1 and WOS and offering a publicly accessible code implementation. Among these, we select HBGL and GACaps-HTC, the two methods reporting the highest macro $F_1$ scores on the RCV1 and WOS datasets, respectively, and test the authors' implementation on our datasets. Additionally, we select other HTC methods that are heavily referenced in the literature, namely, MATCH, HiMatch, and HiAGM. As baselines for comparison, we

also report results on popular flat-classification methods like SVM, BERT, and XML-CNN, additionally testing the latter two with losses optimized for HTC tasks. Despite our best efforts, we were unable to use the existing implementations of other methods that we had initially selected. Firstly, several works have missing dependencies or files that are unspecified in the provided code. Secondly, some works utilize outdated versions of libraries that are no longer available or supported. Lastly, in some cases, there was a lack of adequate instructions or support to extend the work on other datasets. We will briefly touch on the issue of reproducibility in Section 6; as mentioned, to help with this problem, we share publicly both our data splits (when legally possible) and code. In some cases, despite managing to run the method's implementation, we did not achieve good performance on any of our datasets. A more detailed recollection of our experiments on other methods may be found in the supplemental material.

The following paragraphs describe the hyperparameters, preprocessing operations, and settings used with each method we were able to test, as well as a few baselines. For all methods, we perform 3-fold cross-validation on the Bugs dataset to select a common set of hyperparameters. Then, we use these to test each model on all five datasets. A supplementary material document is also provided to further detail the validation procedure.

### 5.2.1. HBGL

We use the public implementation of HBGL [180] and train the model for 960,000 steps, saving the checkpoint with the highest macro $F_1$ score on the validation split, as in the original paper. All hyperparameters excluding the learning rate and batch size (which are selected according to the validation procedure) are set to the same values used by the authors and the number of input tokens to the BERT encoder is set to the maximum number (512).

### 5.2.2. GACaps-HTC

We run the GACaps-HTC [191] implementation on our five datasets, training for a maximum of 200 epochs with early stopping, utilizing the hyperparameters suggested by the authors and tuning the learning rate and batch size. This method also includes an automatic hyperparameter search that tunes parameters based on validation set performance. In line with the authors, we use the "`allenai/scibert_scivocab_uncase`" pre-trained BERT model to generate embeddings for the WOS dataset without fine-tuning and fine-tune a BERT model for all other datasets starting from the "`bert-base-uncased`" dump from HuggingFace. We train with a batch size set to 16 using the Adam optimizer.

### 5.2.3. MATCH

We test the proposal from Zhang et al. [165] on our datasets. Our datasets have no metadata; therefore, we test the model as limited to using the normalization mechanism for the hierarchy of labels. Additionally, we could not reproduce the joint embedding pre-training on our datasets; hence, we replaced it with Word2Vec embeddings, which we previously fine-tuned on our dataset for 20 epochs. Before training Word2Vec embeddings, we preprocess the dataset by removing URLs, hexadecimal codes, and memory addresses (only for the Bugs dataset) and make all words lowercase. The text is then tokenized with NLTK's word tokenizer, and punctuation marks are removed. The model uses a BCE loss with regularization terms optimized using Adam.

### 5.2.4. HiAGM

We use the public implementation of HiAGM [97] and test its performance on our datasets. The text is preprocessed with the same procedure described by its authors, consisting of making all words lowercase and removing special characters, stopwords, and punctuation marks. The Adam optimizer is used during training.

### 5.2.5. BERT

We adopt the pre-trained "`bert-base-cased`" model from the HuggingFace library [34,223]. BERT stacks several Transformer encoder blocks and we concatenate the (un-pooled) "[CLS]" representation from the last three blocks to obtain a more meaningful document representation, as in Marcuzzo et al. [101], Tanaka et al. [224]. As with MATCH, we preprocess the dataset by removing URLs, hexadecimal codes for the Bugs dataset, and lowercase words. For tokenization, we use BERT's Tokenizer, which is derived from WordPiece [26]. The BCE loss is adopted, and AdamW is utilized as an optimizer.

### 5.2.6. XML-CNN

We test a shallow CNN-based model from Liu et al. [225], Adhikari et al. [226]. Word embeddings are initialized with pre-trained GloVe embeddings [22] based on results obtained on the validation set. The text is preprocessed as was previously described for MATCH. Additionally, we remove stopwords from all corpora, since we find this step improves metric results on the validation set. We use the BCE loss with the AdamW optimizer.

### 5.2.7. CHAMP/MATCH Losses

We additionally test the BERT model and XML-CNN with the CHAMP and MATCH loss proposed by Vaswani et al. [107] and Zhang et al. [165]. For CHAMP, we implemented the "hard" version of their loss function. We test these approaches using the same optimization method, as well as the same preprocessing and tokenization steps described for BERT and XML-CNN, respectively.

### 5.2.8. SVM

As a traditional baseline, we test the performance of simple SVM-based approaches with a one-vs-rest strategy. We test a standard flat approach that performs multiclass classification on the most specific label for each example, as well as a naive multilabel approach with no hierarchical information (to simulate an NMLNP). We apply the same preprocessing described for XML-CNN and use a TF-IDF representation to generate word feature vectors. As expected for this type of text representation, the removal of stopwords improves the performance on all datasets. On the Bugs dataset, we perform a more aggressive and targeted cleaning procedure, removing pieces of text that refer to memory addresses and codes, such as to reduce the size of the generated vocabulary.

### 5.3. Results

In this section, we report and discuss the results obtained with each method on the five selected datasets. As previously mentioned, RCV1-v2 and BGC have been shared with pre-defined training and testing splits, allowing easy comparisons with other works. The results reported in the literature on RCV1 are shown in Table 8 (only when using standard splits), together with results on the widely popular WOS-46985. Only works reporting F1 metrics are reported. The BGC dataset is relatively new and only a handful of works have results on it. Conversely, the WOS dataset, as well as the Amazon and Bugs datasets that we collected, do not have standardized splits. For this reason, we decided to share our training and testing splits to allow for better reproducibility. We also report results over these same datasets using a 3-fold cross-validation strategy with both standard and hierarchical metrics (when possible). The latter can be found in Tables 9 and 10, while the results over the individual splits are reported in Table 11.

**Table 8.** Performance reported in the literature on RCV1-v2 and WOS datasets (best results in bold).

| Method | RCV1-v2 * | | WOS ** | |
|---|---|---|---|---|
| | micro-$F_1$ | macro-$F_1$ | micro-$F_1$ | macro-$F_1$ |
| HTrans [148] | 0.805 | 0.585 | - | - |
| NeuralClassifier (RCNN) [149] | 0.810 | 0.533 | - | - |
| HiLAP [99] | 0.833 | 0.601 | - | - |
| HiAGM-TP [97] | 0.840 | 0.634 | 0.858 | 0.803 |
| RLHR [167] | - | - | 0.785 | 0.792 |
| HCSM [168] | 0.858 | 0.609 | **0.921** | 0.807 |
| HiMatch [124] | 0.847 | 0.641 | 0.862 | 0.805 |
| HIDDEN [169] | 0.793 | 0.473 | - | - |
| HE-AGCRCNN [170] | 0.778 | 0.513 | - | - |
| HVHMC [171] | - | - | 0.743 | - |
| SASF [126] | - | - | 0.867 | 0.811 |
| HTCInfoMax [177] | 0.835 | 0.627 | 0.856 | 0.800 |
| PAAM-HiA-T5 [178] | 0.872 | 0.700 | 0.904 | 0.816 |
| HPT [136] | 0.873 | 0.695 | 0.872 | 0.819 |
| HGCLR [91] | 0.865 | 0.683 | 0.871 | 0.812 |
| Seq2Tree [93] | 0.869 | 0.700 | 0.872 | 0.825 |
| HBGL [180] | 0.872 | **0.711** | 0.874 | 0.820 |
| P-tuning v2 (SPP-tuning) [138] | - | - | 0.875 | 0.800 |
| LD-GGNN [186] | 0.842 | 0.641 | 0.851 | 0.805 |
| LSE-HiAGM [123] | 0.839 | 0.646 | 0.860 | 0.800 |
| Seq2Label [121] | 0.874 | 0.706 | 0.873 | 0.819 |
| HTC-CLIP [190] | - | - | 0.879 | 0.816 |
| GACaps [191] | 0.868 | 0.698 | 0.876 | **0.828** |
| HiDEC [194] | 0.855 | 0.651 | - | - |
| UMP-MG [129] | - | - | 0.859 | 0.813 |
| LED [196] | **0.883** | 0.697 | 0.870 | 0.813 |
| (HGCLR-based + aug) [198] | 0.862 | 0.679 | 0.874 | 0.821 |
| K-HTC [135] | - | - | 0.873 | 0.817 |
| HiTIN (BERT) [200] | 0.867 | 0.699 | 0.872 | 0.816 |
| HierVerb (few-shot) [137] | 0.655 | 0.351 | 0.809 | 0.738 |
| DLAC [202] | - | - | 0.868 | 0.810 |
| PeerHTC [130] | - | - | 0.742 | 0.674 |
| HJCL [206] | 0.870 | 0.705 | - | - |

* Refers to RCV1-V2 with split 20,834/2315/781,265 (train/validation/testing). ** Refers to WOS-46985.

**Table 9.** Performance metrics on the test set for each model on the Bugs, WOS, and Amazon datasets, with best results outlined in bold. (CL/ML = CHAMP/MATCH loss). ↑ stands for higher is better, while ↓ stand for lower is better.

| Model | Acc ↑ | $F_1$ ↑ | h-$F_1$ ↑ | AHC ↓ |
|---|---|---|---|---|
| | | Bugs | | |
| MATCH | 0.3624 [±0.0491] | 0.3635 [±0.0559] | 0.3638 [±0.0560] | 0.5501 [±0.0834] |
| HiAGM | 0.4482 [±0.0069] | 0.5218 [±0.0045] | 0.5214 [±0.0046] | 0.7672 [±0.0527] |
| BERT + CL | 0.4822 [±0.0066] | 0.4965 [±0.0078] | 0.4968 [±0.0079] | 0.4047 [±0.0182] |
| BERT + ML | 0.5061 [±0.0099] | 0.5165 [±0.0095] | 0.5172 [±0.0096] | 0.4634 [±0.0238] |
| XML-CNN + CL | 0.2603 [±0.0037] | 0.2515 [±0.0062] | 0.2522 [±0.0247] | **0.2443 [±0.0052]** |
| XML-CNN + ML | 0.2823 [±0.0110] | 0.2767 [±0.0058] | 0.2774 [±0.0057] | 0.2885 [±0.0188] |
| HBGL | **0.5763 [±0.0003]** | **0.5709 [±0.0032]** | **0.5710 [±0.0029]** | 0.6320 [±0.0037] |
| GACaps | 0.4916 [±0.0052] | 0.5430 [±0.0110] | 0.5445 [±0.0110] | 0.6476 [±0.0725] |
| BERT | 0.5070 [±0.0113] | 0.5204 [±0.0093] | 0.5209 [±0.0091] | 0.4606 [±0.0119] |
| XML-CNN | 0.2800 [±0.0130] | 0.2710 [±0.0163] | 0.2716 [±0.0160] | 0.3000 [±0.0239] |
| SVM (MultiL) | 0.3029 [±0.0035] | 0.4187 [±0.0044] | 0.4207 [±0.0043] | 0.5254 [±0.0086] |
| SVM (MultiC) | 0.5496 [±0.0039] | 0.4724 [±0.0061] | - | - |

**Table 9.** *Cont.*

| Model | Acc ↑ | F$_1$ ↑ | h-F$_1$ ↑ | AHC ↓ |
|---|---|---|---|---|
| | | WOS | | |
| MATCH | 0.5932 [±0.0161] | 0.6672 [±0.0145] | 0.6674 [±0.0145] | 0.3891 [±0.0114] |
| HiAGM | 0.6513 [±0.0159] | 0.7544 [±0.0065] | 0.7541 [±0.0066] | 0.4272 [±0.0356] |
| BERT + CL | 0.7647 [±0.0064] | 0.7928 [±0.0066] | 0.7929 [±0.0066] | 0.1941 [±0.0130] |
| BERT + ML | 0.7718 [±0.0027] | 0.7974 [±0.0030] | 0.7974 [±0.0030] | 0.2120 [±0.0104] |
| XML-CNN + CL | 0.4488 [±0.0078] | 0.5408 [±0.0095] | 0.5410 [±0.0095] | **0.1853 [±0.0091]** |
| XML-CNN + ML | 0.4759 [±0.0109] | 0.5688 [±0.0057] | 0.5691 [±0.0057] | 0.2044 [±0.0097] |
| HBGL | **0.8014 [±0.0002]** | **0.8221 [±0.0004]** | **0.8221 [±0.0004]** | 0.2140 [±0.0024] |
| GACaps | 0.7376 [±0.0078] | 0.8063 [±0.0050] | 0.8067 [±0.0050] | 0.2753 [±0.0136] |
| BERT | 0.7712 [±0.0067] | 0.7981 [±0.0059] | 0.7981 [±0.0059] | 0.2087 [±0.0069] |
| XML-CNN | 0.4714 [±0.0115] | 0.5628 [±0.0085] | 0.5630 [±0.0085] | 0.1978 [±0.0105] |
| SVM (MultiL) | 0.5051 [±0.0026] | 0.6611 [±0.0018] | 0.6619 [±0.0018] | 0.2172 [±0.0061] |
| SVM (MultiC) | 0.7609 [±0.0033] | 0.7397 [±0.0045] | - | - |
| | | Amazon | | |
| MATCH | 0.8717 [±0.0087] | 0.9039 [±0.0028] | 0.9039 [±0.0028] | 0.0801 [±0.0028] |
| HiAGM | 0.8735 [±0.0044] | 0.9029 [±0.0030] | 0.9029 [±0.0030] | 0.0858 [±0.0037] |
| BERT + CL | 0.8912 [±0.0019] | 0.9192 [±0.0015] | 0.9192 [±0.0015] | 0.0632 [±0.0016] |
| BERT + ML | **0.8960 [±0.0021]** | **0.9214 [±0.0015]** | **0.9214 [±0.0015]** | 0.0658 [±0.0020] |
| XML-CNN + CL | 0.8242 [±0.0038] | 0.8849 [±0.0018] | 0.8850 [±0.0018] | 0.0709 [±0.0015] |
| XML-CNN + ML | 0.8279 [±0.0036] | 0.8860 [±0.0021] | 0.8860 [±0.0021] | 0.0782 [±0.0011] |
| HBGL | 0.8405 [±0.0002] | 0.8796 [±0.0012] | 0.8796 [±0.0012] | 0.1134 [±0.0000] |
| GACaps | 0.8673 [±0.0019] | 0.9058 [±0.0010] | 0.9057 [±0.0011] | 0.0701 [±0.0005] |
| BERT | 0.8954 [±0.0020] | 0.9212 [±0.0014] | 0.9212 [±0.0014] | 0.0661 [±0.0008] |
| XML-CNN | 0.8290 [±0.0030] | 0.8867 [±0.0016] | 0.8867 [±0.0017] | 0.0774 [±0.0020] |
| SVM (MultiL) | 0.7340 [±0.0016] | 0.8491 [±0.0010] | 0.8492 [±0.0010] | **0.0628 [±0.0010]** |
| SVM (MultiC) | 0.8666 [±0.0007] | 0.8677 [±0.0007] | - | - |

The standard deviation over the 2 repetitions of 3-fold cross-validation is reported in brackets.

**Table 10.** Performance metrics on the test set for the BGC and RCV1-v2 datasets, with best results outlined in bold. ↑ stands for higher is better, while ↓ stand for lower is better.

| Model | Acc ↑ | F$_1$ ↑ | h-F$_1$ ↑ | AHC ↓ |
|---|---|---|---|---|
| | | RCV1-v2 | | |
| MATCH | 0.5149 [±0.0010] | 0.5308 [±0.0056] | 0.5309 [±0.0056] | 0.3478 [±0.0005] |
| HiAGM | 0.6035 [±0.0064] | 0.6836 [±0.0075] | 0.6830 [±0.0083] | 0.3236 [±0.0001] |
| BERT + CL | 0.6409 [±0.0004] | 0.6612 [±0.0127] | 0.6613 [±0.0125] | 0.1929 [±0.0304] |
| BERT + ML | 0.6383 [±0.0059] | 0.6805 [±0.0073] | 0.6806 [±0.0073] | 0.2270 [±0.0076] |
| XML-CNN + CL | 0.5449 [±0.0099] | 0.4898 [±0.0058] | 0.4905 [±0.0055] | **0.1768 [±0.0121]** |
| XML-CNN + ML | 0.5460 [±0.0089] | 0.4832 [±0.0161] | 0.4840 [±0.0164] | 0.1917 [±0.0156] |
| HBGL | 0.6588 [±0.0015] | **0.7284 [±0.0001]** | **0.7285 [±0.0002]** | 0.2312 [±0.0127] |
| GACaps | 0.6390 [±0.0012] | 0.7103 [±0.0060] | 0.7090 [±0.0057] | 0.2629 [±0.0118] |
| BERT | 0.6391 [±0.0036] | 0.6722 [±0.0305] | 0.6723 [±0.0304] | 0.1959 [±0.0480] |
| XML-CNN | 0.5516 [±0.0023] | 0.4923 [±0.0124] | 0.4932 [±0.0127] | 0.1815 [±0.0098] |
| SVM (MultiL) ** | 0.4971 | 0.5456 | 0.5472 | 0.2340 |
| SVM (MultiC) ** | **0.7289** | 0.4416 | - | - |
| | | BGC | | |
| MATCH | 0.3876 [±0.0009] | 0.4800 [±0.0015] | 0.4802 [±0.0012] | 0.4793 [±0.0059] |
| HiAGM | 0.4112 [±0.0055] | 0.5483 [±0.0030] | 0.5484 [±0.0024] | 0.5483 [±0.0144] |
| BERT + CL | 0.4674 [±0.0008] | 0.6058 [±0.0156] | 0.6060 [±0.0155] | 0.3303 [±0.0017] |
| BERT + ML | 0.4740 [±0.0004] | 0.6116 [±0.0049] | 0.6119 [±0.3450] | 0.3450 [±0.0017] |
| XML-CNN + CL | 0.3544 [±0.0012] | 0.3870 [±0.0029] | 0.3873 [±0.0028] | 0.3054 [±0.0121] |
| XML-CNN + ML | 0.3549 [±0.0018] | 0.3798 [±0.0076] | 0.3803 [±0.0077] | **0.2875 [±0.0227]** |

**Table 10.** *Cont.*

| Model | Acc ↑ | F$_1$ ↑ | h-F$_1$ ↑ | AHC ↓ |
|---|---|---|---|---|
| HBGL | 0.5060 [±0.0006] | **0.6782 [±0.0016]** | **0.6779 [±0.0015]** | 0.3540 [±0.0032] |
| GACaps | 0.4527 [±0.0071] | 0.6459 [±0.0059] | 0.6446 [±0.0063] | 0.4517 [±0.0150] |
| BERT | 0.4711 [±0.0032] | 0.6100 [±0.0119] | 0.6102 [±0.0116] | 0.3490 [±0.0334] |
| XML-CNN | 0.3602 [±0.0028] | 0.3996 [±0.0092] | 0.4000 [±0.0091] | 0.3096 [±0.0073] |
| SVM (MultiL) ** | 0.3495 | 0.5129 | 0.5148 | 0.3985 |
| SVM (MultiC) ** | **0.6285** | 0.2792 | - | - |

The standard deviation over 2 runs on the same standardized splits is reported in brackets. ** SVMs run on fixed splits have deterministic results if run for enough iterations.

**Table 11.** F$_1$ score (macro) results on one split of each dataset (average of 2 runs). Best results are outlined in bold.

| Method | RCV1-v2 | WOS | Amz | BGC | Bugs |
|---|---|---|---|---|---|
| MATCH | 0.5308 | 0.6719 | 0.9048 | 0.4800 | 0.4064 |
| HiAGM | 0.6836 | 0.7484 | 0.9041 | 0.5483 | 0.5263 |
| BERT + CL | 0.6612 | 0.7958 | 0.9200 | 0.6058 | 0.5046 |
| BERT + ML | 0.6805 | 0.8007 | **0.9214** | 0.6116 | 0.5218 |
| XML-CNN + CL | 0.4898 | 0.5471 | 0.8856 | 0.3870 | 0.2517 |
| XML-CNN + ML | 0.4832 | 0.5645 | 0.8863 | 0.3798 | 0.2784 |
| HBGL | **0.7284** | **0.8221** | 0.8794 | **0.6781** | **0.5710** |
| GACaps | 0.7095 | 0.8064 | 0.9058 | 0.6464 | 0.5433 |
| BERT | 0.6722 | 0.7991 | 0.9206 | 0.6100 | 0.5235 |
| XML-CNN | 0.4923 | 0.5631 | 0.8855 | 0.3996 | 0.2802 |
| SVM (MultiL) | 0.5456 | 0.6610 | 0.8484 | 0.5129 | 0.4184 |
| SVM (MultiC) | 0.4416 | 0.7438 | 0.8676 | 0.2792 | 0.4700 |

### 5.3.1. Comparison

On the Linux Bugs datasets, three methods performed particularly well: HBGL, GACaps, and HiAGM. In terms of accuracy, HBGL is the best method, with the simplified SVM with multiclass objective coming second. However, the latter is much worse than the hierarchical methods in terms of F$_1$ score. This traditional, non-hierarchical approach using TF-IDF embeddings and an SVM classifier is likely to be effective on the Linux Bugs dataset because of how technical and noisy its language is: the heavier preprocessing steps to produce the TF-IDF vocabulary can filter out some symbols and noisy pieces of text that are not relevant to the classification. Conversely, for context-aware methods that have been pre-trained on longer and less technical texts, such as BERT, this may come as a disadvantage. Despite this, the latter model achieves good results, but it is plausible to assume that it suffers from the lack of structure in the bodies of text within the dataset. GACaps performed significantly worse than HBGL in terms of accuracy and slightly worse in terms of F$_1$ score but also improved over BERT and HiAGM with a score in line with the BERT + ML model. HiAGM performed similarly to the BERT + ML model, although its hierarchy-aware mechanism significantly favors the F$_1$ score, indicating more balanced results across different categories. MATCH, XML-CNN, and the two hierarchical losses provided worse or inconsequential results. Overall, the performance comparison seems to suggest that integrating hierarchical information is not straightforward but can be advantageous (such as in the case of HBGL, GACaps, and HiAGM).

While HBGL and GACaps remained the two top-performing methods, the Web of Science dataset largely favors the BERT-based approaches over HiAGM. The improvement provided by the MATCH loss is negligible, while the CHAMP loss seems to provide worse results (as was the case for the previous dataset). In general, then, the good results can be attributed to the semantic interpretation capabilities of BERT, rather than the hierarchical information that the former loss could provide. In this case, the top three methods also surpassed the accuracy of traditional approaches based on SVMs, most likely because the

text is much more structured and less noisy. HiAGM still provided decent results, though HBGL scores $\sim 9\%$ higher on both metrics.

The Amazon $5 \times 5$ dataset synthesized for this work proved to be a much easier task for all the models, which may be attributed to the fact that the hierarchy is very well-structured and the fact that the samples have been devised to be very balanced. Notably, most methods on other datasets would often misclassify the entirety of some low-frequency classes during testing. This happened on all datasets except for Amazon $5 \times 5$, which, in that respect, is much "easier" because there are no real low-frequency classes. Interestingly, all neural approaches except HBGL and XML-CNN (which scored lower) show comparable performance, with BERT + ML achieving the highest scores. BERT performed very well, with a very minor improvement when adding MATCH's loss to its optimization. BERT's context understanding capabilities can likely be fully exploited when training on this balanced dataset with long pieces of text. Indeed, this is the dataset with the longest average example length. Considering that HBGL and GACaps surpassed the other methods on most datasets, we can conclude that their hierarchy injection strategy is particularly beneficial in datasets with a deeper and more imbalanced hierarchy.

Among those discussed so far, the RCV1-v2 corpus is the first to have a deeper and more complex hierarchy. HBGL performed best in terms of $F_1$ metrics, followed by GACaps and HiAGM, further validating the injection of hierarchical information in HTC scenarios, as opposed to flat classification. However, the best accuracy was achieved by the multiclass SVM, though it should be taken with a grain of salt because of how strong the flattening assumptions are on the relatively complex hierarchy. Indeed, this is reflected in its much lower $F_1$ score, resulting from the limited ability to correctly predict minority classes. All the BERT variants performed closely on this dataset, with the MATCH loss strategy giving slightly better metrics.

Lastly, the Blurb Genre Collection also proved quite difficult to categorize. Again, HBGL and GACaps obtained the best $F_1$ score, followed by BERT-based models. Accuracy was yet again dominated by the multiclass SVM approach, though with a much worse $F_1$ score—a symptom of severe imbalance, like in the previous dataset. Finally, HiAGM performs significantly worse than the best models and even lags behind BERT-based approaches on this dataset.

Overall, HBGL emerged as the most promising method, outperforming the rest in terms of $F_1$ score and accuracy across the majority of datasets. GACaps and BERT + ML also exhibited good performance, with the former outperforming the latter in most cases, but with some instances where results were close. The hierarchy-aware regularization strategies referred to as CHAMP and MATCH did not appear to consistently influence results. At times, they do provide a marginal boost in the precision or recall over the flattened counterpart, but the advantage of using them is not evident when compared to BERT's base model. (Full results that include precision and recall, as well as their hierarchical counterparts, are available as Supplementary Materials). The CNN-based model (XML-CNN) scored much lower than BERT across all datasets, and our experimental approach that integrated the aforementioned regularization strategies within its framework did not provide performance boosts.

Unfortunately, the evaluation based on hierarchical $F_1$ score did not provide any further insight into the benefits of the MATCH/CHAMP regularization. Concerning the AHC, the XML-CNN had generally lower (better) results. However, we noticed that this outcome is associated with the model's relatively high precision and low recall, indicating a tendency to predict only a few labels accurately. The AHC appeared to favor this behavior, since it is negatively impacted by false positives and not by false negatives. In a model with high precision and low recall, false positives tend to be rare. This was also observed in several SVM runs that resulted in low AHC. As a consequence, this metric should only be considered jointly with other performance indicators, where it can be used as a measure of the "magnitude" of the misclassifications made by the models. Among the models with less imbalance between precision and recall, HBGL generally made less severe mistakes

than GACaps and HiAGM on all datasets, excluding Amazon. Indeed, on the Amazon dataset, GACaps and HiAGM outperformed HBGL across all metrics, AHC included.

Table 11 contains the results on one split of each dataset to enable easier comparisons with existing and future works that measure performance on the same data. Briefly, HBGL and GACaps showed the best results in terms of macro $F_1$ score, as expected.

5.3.2. Inference Time

In Table 12, we report the time required during inference for each model using a single example (i.e., batch size set to 1). Tests were carried out on an NVIDIA RTX 2080Ti GPU for all models that allow it. SVM-based models were run on a CPU on a separate machine running an Intel i7-8700. The machines had 64 and 48 GB of RAM, respectively.

The results in the table follow an interesting trend. The fastest method is MATCH, followed closely by the SVM-based and XML-CNN approaches. HiAGM is slightly slower, though not by a large magnitude. As expected from large neural models, HBGL and GACaps have the longest inference time; both methods use a BERT model to produce contextualized embeddings, resulting in a longer inference time than the flat BERT model. Both these methods are more than two orders of magnitude slower than HiAGM.

Given these results and the difference in performance metrics, a real-time system would likely decide between the fast yet still reliable results of HiAGM, the slower BERT-based model, and the best-performing but slowest HBGL. The choice comes down to the speed requirements of the real-time system, but we can consider BERT (and variants) to be a good compromise between performance and speed. Moreover, if the specific system has to deal with well-structured (syntactically and semantically) documents with simpler hierarchies, models like BERT are likely to be the better choice. Conversely, the hierarchical injection strategy adopted by HBGL appears to significantly boost performance when dealing with more complex hierarchies and noisy input, albeit with a noticeable increase in inference time.

**Table 12.** Average of inference times on the Bugs dataset, averaged over all runs. Model variants are also averaged, as their inference time is comparable.

| Method | Time (ms) |
| --- | --- |
| MATCH | $8.55 \times 10^{-3}$ |
| HiAGM | $2.13 \times 10^{-1}$ |
| SVM | $2.68 \times 10^{-2}$ |
| BERT | $1.13 \times 10^{1}$ |
| XML-CNN | $1.09 \times 10^{-2}$ |
| HBGL | $7.38 \times 10^{1}$ |
| GACaps-HTC | $2.67 \times 10^{1}$ |

*5.4. Discussion*

In the last few sections, we outlined the best-performing methods, those with faster inference time, and potential trade-offs. In this section, we summarize our views on the experimental part of this survey.

Overall, global attention-based hierarchical models like HBGL and GACaps showcased the best performance, even on the most difficult datasets. Both build upon the Transformer-based BERT, fine-tuning the pre-trained LM to learn contextualized representations and adding hierarchical constraints to make use of the hierarchical knowledge. The most notable downside of these approaches is their computational cost: training these models is by far the most expensive process among the tested methods, with inference time following a similar trend. Hence, their usage on real-time systems could be problematic, depending on the amount of data that have to be processed simultaneously. Both HBGL and GACaps were straightforward to implement on our datasets and performed in line with what was advertised in the respective research articles. Fine-tuning BERT models also proved to be a valid choice in terms of performance and ease of use, particularly in datasets with simple

hierarchies and more structured texts. Traditional, SVM-based approaches applied naively did not perform well on datasets with complex and deep hierarchies, where proper HTC methods have a significant advantage. However, though not generally the best choice, they can be used to build more refined hierarchical approaches, such as local per-level classifiers. There are many references in the literature on such approaches, yet we did not find a widespread and well-established implementation to use as a reference. Still, for systems that have to deal with noisy text, they are valuable assets, as well as being generally easy to apply. Similar considerations can be made for the CNN-based model we tested.

HiAGM, the reference baseline for many of the works we analyzed, performs well, especially considering its slimmer nature and the faster training time when compared to HBGL or BERT. In terms of usability, it was relatively straightforward to adapt, though some changes had to be made for it to apply to different hierarchies than those devised in the original work.

Things were more complex for MATCH, as much of the training process was deeply embedded in the provided implementation and had to be adapted to our datasets. This was the case for many other methods, and, unfortunately, we were unable to test them because of inherent technical issues within them, which are summarized in the Supplementary Materials.

Hierarchical metrics can be thought of as corrected variants of the original ones, accounting for the hierarchy and disregarding consistency errors that can be easily rectified by considering the hierarchy tree. As such, they are useful for quantifying the "irreparable" mistakes made by methods. However, in our measurements, these did not significantly differ from the standard performance metrics. We do acknowledge, though, that these metrics can be valuable in settings with deeper hierarchies. Concerning AHC, this metric indicates how far off the predictions made by the models are in terms of distance in the hierarchy tree. Our tests revealed that, among others, HBGL performed better than GACaps in this regard. However, this indicator must be considered carefully along with precision and recall, as it is sensitive to the imbalance between these metrics. Still, the idea of penalizing "better mistakes" differently is interesting and warrants further investigation in the future.

## 6. Future Work and Research Directions

Several current challenges are being tackled by current HTC research, providing interesting research directions to explore. In this section, we briefly outline them.

First, a considerable number of works on HTC have recently been taking into consideration the semantics of labels to improve classification performance [110,123,124,171,179]. Some of them devise a way to obtain label embeddings and then use this information to produce label-aware document embeddings. This is different from standard classification approaches where document features are used to compute per-class probabilities, but an interesting direction being explored is the combination of the two. Some proposals even use contextualized LMs to obtain embeddings for few-word labels. Despite this approach producing good results in [124], the usage of contextualized models to produce embeddings of small fragments of text without enough "context" is not completely justified according to the previous literature [1], or at least it does not produce a proper semantic embedding. In a similar vein, some methods attempt to compensate for the lack of external metadata describing the label semantics by computing latent representation using auto-encoders [179] and GCNs [123,170,171].

Second, some methods are focused on reducing the number of trainable parameters, thereby decreasing the demand for computational resources. Techniques like knowledge distillation [167] and prompt-tuning [136–138] are gaining traction in HTC, as we have highlighted in previous sections. Furthermore, notable advancements have recently been achieved in neural architecture search, a technique that automates the design of efficient neural networks to solve a variety of tasks. Although yet to be specifically applied to HTC, it is a promising research direction, considering its recent successes in general NLP and TC problems [227–229]. This also ties in with the enormous impact foundation LLMs

have had on NLP research. Researchers have been able to utilize their few- and zero-shot capabilities to obtain outstanding results on a wide range of NLP tasks with little to no training data. However, such models still struggle in certain scenarios, such as HTC, because of the often inherently long-tailed distribution of labels in such tasks [230] (a characteristic that is very common in real-world scenarios, such as legal document concept labeling). At the same time, tuning the parameters of these models is extremely costly because of the sheer amount of parameters they are composed of [38]. Works such as the one by Bhambhoria et al. [48] propose a solution to these limitations by enhancing entailment-contradiction predictors with LLMs, which allows them to effectively tackle long-tailed distributions without performing resource-intensive parameter updates in very large LLMs.

Another worthwhile topic is that of reproducibility, benchmarks, and metrics. This is often a subject of debate in many sub-domains of machine learning, and rightfully so. First, HTC lacks a proper set of benchmarks to allow for simpler comparison of methods and establishment of the state of the art. The wider TC branch of NLP, for instance, has multiple well-established benchmarks, such as GLUE [231] and SuperGLUE [232]. As we outlined in Section 4.4, there is currently a wide range of datasets being utilized without much consistency. Even the most popular dataset (RCV1) is sometimes used inconsistently (not all the literature utilizes the same split size). Overall, this makes reproducibility and comparability across methods much more difficult than they should be.

While we showcased the excellent efforts by many authors in the creation of hierarchical metrics [15,17,18,105], we also hinted at their flaws, as well as their lack of widespread adoption. Therefore, this is also still an open area of research. Lastly, as we showcased in this work, many proposed methods do not provide enough instructions to make their methods reproducible. Of the methods in Tables 1, 2–4, only about half provide a code implementation, and many of those still end up not being usable on our datasets because of a lack of instructions, missing dependencies, or excessive hardship in adapting it to other data. At the very least, a well-written set of instructions would be sufficient to provide enough information to any practitioner with the intent of reproducing a method.

Lastly, a topic that is drawing more and more attention across all fields of AI is that of *explainability*. Authors are exploring explainability in the broader NLP area by investigating LMs with tools such as Language Interpretability Tool [233], Errudite [234], and iSEA [235]. This type of information could be utilized and refined to understand how an HTC method chooses its path through the hierarchy and how it behaves when adjusting the input with small perturbations. Indeed, exploring the behavior of the model on different branches of the taxonomy could reveal interesting insights into the overall decision process.

## 7. Conclusions

In this article, we provide an overview of HTC approaches and the required background to fully grasp them, as well as an exploration of recent proposals in the field and the broader NLP domain. We describe common frameworks utilized for HTC, as well as their strengths and weaknesses. We also present commonly utilized evaluation metrics, including those that are specifically designed to take into account the label taxonomy in their calculation. We then collect several works and proposals from recent years and perform a more in-depth study on a selection of them, restricting ourselves to those methods that provide results on the most common datasets. Moreover, we provide an overview of commonly utilized datasets in HTC research. On the experimental side, we find that most of the methods analyzed do not provide an implementation, and, unfortunately, even many of those that do are not easily adjustable to different datasets, vastly limiting the practical value of these works. We measured the performance of those methods we managed to reproduce using both traditional and hierarchical metrics and compared them with a set of baselines. Overall, our results suggest that integrating hierarchical information within the classification in HTC datasets can be beneficial, significantly surpassing flat classification with LMs on all but one dataset. While pre-trained LMs remain competitive on datasets with shallow or simple hierarchies, their combination with global hierarchical

approaches displays a clear superiority, as proven by the two most recent methods we tested. Finally, the code from our experiments is released, and our datasets are made available to researchers.

**Supplementary Materials:** The following supporting information can be downloaded at: https://www.mdpi.com/article/10.3390/electronics13071199/s1. Reference [236] is cited in the supplementary materials.

**Author Contributions:** Conceptualization, A.Z. and M.M.; Methodology, A.Z. and M.M.; Software, A.Z. and M.M.; Validation, L.G., M.R., A.Z. and M.M.; Formal analysis, M.M.; Investigation, A.Z. and M.M.; Resources, A.Z.; Data curation, A.Z.; Writing—original draft preparation, A.Z. and M.M.; Writing—Review and editing, A.Z., M.M. and M.R.; Visualization, A.Z. and M.M.; Supervision, L.G., A.G. and A.A.; Project administration, A.G. and A.A.; Funding acquisition, A.A. All authors have read and agreed to the published version of the manuscript.

**Funding:** This study was carried out within the "Interconnected Nord-Est Innovation Ecosystem (iNEST)" project and received funding from the European Union Next-GenerationEU—National Recovery and Resilience Plan (NRRP)—MISSION 4 COMPONENT 2, INVESTIMENT N. ECS00000043 —CUP N. H43C22000540006. This manuscript reflects only the authors' views and opinions; neither the European Union nor the European Commission can be considered responsible for them. This paper was funded by Veneto Agricoltura within the scope of the project "Guaranteeing the continuity of the agrifood chain: the digitization of wholesale markets".

**Data Availability Statement:** All the code produced can be found here: https://gitlab.com/distration/dsi-nlp-publib/-/tree/main/htc-survey-24 (accessed on 6 February 2024). Dataset splits for the publicly available datasets are available on Zenodo: https://doi.org/10.5281/zenodo.7319519 (accessed on 17 March 2024).

**Acknowledgments:** The authors would like to thank NIST for allowing us to utilize the RCV1 dataset in our experiments.

**Conflicts of Interest:** The authors declare no conflicts of interest.

## Abbreviations

The following abbreviations are used in this manuscript:

| | |
|---|---|
| TC | Text Classification |
| NLP | Natural Language Processing |
| HTC | Hierarchical Text Classification |
| HMC | Hierarchical Multilabel Classification |
| BoW | Bag of Words |
| TF-IDF | Term Frequency - Inverse Document Frequency |
| RNN | Recurrent Neural Network |
| CNN | Convolutional Neural Network |
| BERT | Bidirectional Encoder Representations from Transformers |
| GPT | Generative Pre-trained Transformer |
| MLM | Masked Language Modeling |
| LSTM | Long Short-Term Memory |
| GRU | Gated Recurrent Unit |
| MHA | Multi-head Attention |
| CN | Capsule Network |
| DAG | Directed Acyclic Graph |
| (N)MLNP | (Non-)Mandatory Leaf Node Prediction |
| LCA | Lowest Common Ancestor |
| NDCG | Normalized Discounted Cumulative Gain |
| RCV1 | Reuters Corpus-V1 |
| WOS | Web Of Science |
| BGC | Blurb Genre Collection |

| 20NG | 20 NewsGroups |
|------|---------------|
| AAPD | ArXiv Academic Paper dataset |
| BCE | Binary Cross Entropy |
| KG | Knowledge Graph |

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
