# Peer review of "Hierarchical Text Classification and Its Foundations: A Review of Current Research"

_electronics, doi:10.3390/electronics13071199_

Round 1

Reviewer 1 Report

Comments and Suggestions for Authors

A brief summary

This article reviews current research on Hierarchical Text Classification (HTC), emphasizing the unique approach of utilizing hierarchically structured labels to enhance classification performance. It starts by defining HTC and its importance within text classification, then discusses traditional and recently proposed methods, highlighting their contributions. Additionally, the article examines commonly used datasets and the significance of hierarchical evaluation metrics. A distinctive aspect of this work is its comprehensive analysis of recent HTC research trends (2019-2023), including a systematic review of publications, an experimental comparison of recent proposals against non-hierarchical baselines across multiple datasets, and the release of code and datasets for future research.

This paper is very well-prepared.

Specific comments:

Please cite the following paper:

Liu, R., Liang, W., Luo, W., Song, Y., Zhang, H., Xu, R., Li, Y., & Liu, M. (2023). Recent Advances in Hierarchical Multi-label Text Classification: A Survey. ArXiv, abs/2307.16265.

Defiyanti, S., Winarko, E., & Priyanta, S. (2019). A Survey of Hierarchical Classification Algorithms with Big-Bang Approach. 2019 5th International Conference on Science and Technology (ICST), 1, 1-6.

Bhambhoria, R., Chen, L., & Zhu, X. (2023). A Simple and Effective Framework for Strict Zero-Shot Hierarchical Classification. ACL2023.pp. 1782-1792.

Currently, the culmination of deep learning technology is Large Language Models (LLMs). The paper, "A Simple and Effective Framework for Strict Zero-Shot Hierarchical Classification," attempts in detail to use LLMs for Zero-Shot Hierarchical Classification. Please elaborate more on leveraging LLM for Hierarchical Classification in Section 2.1.4.

There are instances where the "https://doi.org/" appears twice consecutively in some references, which might be a bibtex error, for example, papers [60], [71], [116], [185], [189], [214], etc.

Please verify the Methods listed in the table that are mentioned only by reference links without names. For instance, in Table 2, the paper [106] "A Hierarchical Fine-Tuning Approach Based on Joint Embedding of Words and Parent Categories for Hierarchical Multi-label Text Classification" does not have a method name listed, but in fact, the method is HFT-ONLSTM (https://github.com/masterzjp/HFT-ONLSTM).

Comments on the Quality of English Language

Minor editing of English language required.

Reviewer 2 Report

Comments and Suggestions for Authors

Dear Authors, dear Editor,

In the manuscript, the problem of Hierarchical Text Classification analysis is discussed. First, the main concepts, definitions, and the current state of the problem are summarized. The paper provides an analysis of common approaches to this paradigm and its evaluation measures, explores the NLP background of text representation, and examines various neural architectures. Additionally, a review of a variety of recent (2019-2023) common frameworks for NLP research is presented. The second part of the research is practical/experimental, where 5 different datasets are analyzed using different methods, and short results and conclusions are provided for each. Future work and research directions are discussed at the end, confirming that the problem and methods are still under development, especially addressing issues such as reproducibility, benchmarks, and metrics. The work demonstrates that many proposed methods do not provide sufficient instructions to ensure the reproducibility of their approaches.

The manuscript is structured as a review article but incorporates a experimental/practical part. In the supplementary materials the detailed training results for various methods are provided. A brief overview and definition of optimal training parameters are included for each method. Additionally, the second part discussing 'Issues with other tested methods' describes the challenges and limitations defined during the research. The GitHub repository has been released and datasets are made available for further analysis.

With minor corrections I recommend accepting the paper for publication in the journal.

The supplementary materials could be enhanced by addressing the following points. 

  1. I suggest beginning the supplementary materials with a sentence that highlights and links to the page where all training datasets are collected, such as https://zenodo.org/records/7319519 and/or https://gitlab.com/distration/dsi-nlp-publib/-/tree/main/htc-survey-24/data/taxonomies. Also, include a reference to the GitHub repository at https://gitlab.com/distration/dsi-nlp-publib/-/tree/main/htc-survey-24, along with any other related or useful links.

  2. Please add the missing reference links to these methods in the supplementary materials. The citations [2-6] for classifiers that do not work are included. To keep consistency please add references for other methods (e.g., RLHR, BERD) that are mentioned in the paper.

  3. There is a discrepancy between the mention of only three datasets at https://zenodo.org/records/7319519 and the assertion of more datasets being analyzed in the supplementary materials. Additionally, no links to the training datasets are provided. To address this, include references to all analyzed datasets and provide links to the training datasets used, such as the Blurb Genre Collection and RCV1-v2.

Reviewer 3 Report

Comments and Suggestions for Authors

The manuscript provides a comprehensive review of Hierarchical Text Classification (HTC), but could you elaborate on how the recently proposed HTC methods compare against traditional non-hierarchical methods in terms of scalability and adaptability to new, unseen data?

In the experiments and analysis section, it's mentioned that five datasets were used for benchmarking. Can you provide more details on the criteria for selecting these specific datasets and how they represent the diversity of applications in HTC?

The manuscript discusses the importance of evaluation metrics tailored to hierarchical settings. Could you explore further how these hierarchical metrics offer advantages over standard metrics in assessing the performance of HTC models, particularly in real-world applications?

Regarding future work and research directions, the author outlines areas for further exploration. Could you discuss potential interdisciplinary applications of HTC, especially in emerging fields like biomedical informatics or legal document analysis, where hierarchical classification could be particularly beneficial?

The author mentions the release of code and dataset splits for public use. Can you provide insight into the challenges faced during the implementation of HTC methods and how the released resources can help mitigate these challenges for future research?

Comments on the Quality of English Language

Moderate editing of English language required

Reviewer 4 Report

Comments and Suggestions for Authors

The article "Hierarchical Text Classification and its Foundations: a Review of Current Research" provides a comprehensive and insightful overview of the advancements in Hierarchical Text Classification (HTC). It skillfully bridges the gap between traditional text classification methods and the nuanced demands of hierarchical label structures, showcasing the authors' deep understanding of both the theoretical underpinnings and practical applications. By offering a detailed analysis of recent methodologies, datasets, and evaluation metrics, this work stands as a valuable resource for researchers and practitioners alike, pushing the boundaries of NLP research towards more sophisticated and accurate text classification systems.

Round 2

Reviewer 3 Report

Comments and Suggestions for Authors

Authors have addressed all the concerns. The manuscript looks good.

Comments on the Quality of English Language

Minor editing of English language required